# Molecular pathway analysis towards understanding tissue vulnerability in spinocerebellar ataxia type 1

Terri M Driessen[1], Paul J Lee[1], Janghoo Lim[1,2,3]*

[1]Department of Genetics, Yale School of Medicine, New Haven, Unites States; [2]Department of Neuroscience, Yale School of Medicine, New Haven, Unites States; [3]Program in Cellular Neuroscience, Neurodegeneration and Repair, Yale School of Medicine, New Haven, Unites States

**Abstract** The neurodegenerative disorder spinocerebellar ataxia type 1 (SCA1) affects the cerebellum and inferior olive, though previous research has focused primarily on the cerebellum. As a result, it is unknown what molecular alterations are present in the inferior olive, and whether these changes are found in other affected tissues. This study addresses these questions for the first time using two different SCA1 mouse models. We found that differentially regulated genes in the inferior olive segregated into several biological pathways. Comparison of the inferior olive and cerebellum demonstrates that vulnerable tissues in SCA1 are not uniform in their gene expression changes, and express largely discrete but some commonly enriched biological pathways. Importantly, we also found that brain-region-specific differences occur early in disease initiation and progression, and they are shared across the two mouse models of SCA1. This suggests different mechanisms of degeneration at work in the inferior olive and cerebellum.
DOI: https://doi.org/10.7554/eLife.39981.001

*For correspondence:
janghoo.lim@yale.edu

Competing interests: The authors declare that no competing interests exist.

## Introduction

A hallmark of neurodegenerative diseases is brain-region-specific cell death and dysfunction. Why specific brain regions are affected and others remain unscathed is a key question in the field. In order to answer this question, the innate differences and similarities in affected tissues must first be established. Studies in a diverse subset of neurodegenerative disorders have begun using large-scale transcriptomics and proteomics to assess brain region-dependent molecular alterations. For example, a large RNA-sequencing (RNA-seq) study conducted across brain regions and peripheral tissues in Huntington's disease (HD) mouse models identified gene clusters related to transcription, chromatin factors, and mitochondria that appear to be specific to the striatum (*Langfelder et al., 2016*). The same study also found glia-related genes that were significantly enriched in the striatum and cortex, as well as other tissues thought to be less affected in HD, such as the cerebellum and brainstem (*Langfelder et al., 2016*). In Alzheimer's Disease (AD), regional variability in transcriptional changes were observed early in disease progression, but the similarities between tissues increased as animals aged (*Landel et al., 2014*). Finally, a recent study in spinocerebellar ataxia type 3 (SCA3) identified brain region-dependent variability in gene expression changes prior to the onset of motor phenotypes despite modest transcriptional alterations in brain regions that remain largely unscathed (*Toonen et al., 2018*). Collectively, this suggests that a subset of gene clusters spans affected tissues regardless of the degree of tissue vulnerability, while other genes may be enriched in discrete affected tissues. Disease progression may also play a role in dictating the degree of overlap between two affected tissues, with a higher degree of variability between tissues early on in disease.

Mouse models for neurodegenerative disorders can be utilized to query the underlying differences and similarities in affected tissues, including the models generated for spinocerebellar ataxia type 1 (SCA1). SCA1 is a fatal late-onset neurodegenerative disorder that is characterized by impaired motor coordination and balance (*Globas et al., 2008*). Atrophy and loss of Purkinje cells (PCs) in the cerebellum and cell death in the inferior olive have been identified in postmortem human tissue (*Koeppen, 2005*; *Koeppen et al., 2013*). Many of these behavioral and pathological traits have been faithfully recaptured in both a transgenic, and a knockin, mouse model for SCA1 (*Burright et al., 1995*; *Clark et al., 1997*; *Watase et al., 2002*). Both SCA1 models demonstrate motor incoordination and PC pathology, and develop hallmark nuclear inclusions in PCs (*Burright et al., 1995*; *Clark et al., 1997*; *Watase et al., 2002*).

SCA1 is caused by a polyglutamine (polyQ) expansion in the ATAXIN-1 (ATXN1) protein, which is involved in gene regulation, including transcription (*Orr et al., 1993*). Transcriptional regulators, including Capicua, SMRT, and RORa/Tip-60, have been shown to interact with ATXN1 (*Fryer et al., 2011*; *Gehrking et al., 2011*; *Kim et al., 2013b*; *Lam et al., 2006*; *Lim et al., 2008*; *Serra et al., 2006*; *Tsai et al., 2004*). Since ATXN1 functions as a transcriptional regulator, several previous studies have investigated how polyQ-expanded ATXN1 can impact the normal transcriptional signature of the cerebellum (*Crespo-Barreto et al., 2010*; *Cvetanovic et al., 2011*; *Gatchel et al., 2008*; *Ingram et al., 2016*; *Lin et al., 2000*; *Serra et al., 2004*). These studies have identified that various biological pathways, including glutamate signaling, calcium signaling, and long-term depression, are enriched in this tissue at different time-points (*Crespo-Barreto et al., 2010*; *Gatchel et al., 2008*; *Ingram et al., 2016*; *Serra et al., 2004*). Despite the prevalence of transcriptomic data collected from the SCA1 cerebellum, other affected regions, such as the inferior olive, have remained largely unstudied.

The cerebellum and inferior olive are intricately connected. Neurons of the inferior olive, a collection of subnuclei in the rostral medulla, send climbing fibers to innervate PCs (*Bengtsson and Hesslow, 2006*). PC axons, in turn, synapse onto deep cerebellar nuclei, which send efferent fibers to various brain regions including the inferior olive. Previous studies have found inferior olive cell loss in postmortem SCA1 human tissue, as well as changes in climbing fiber synaptic density on PCs (*Koeppen et al., 2013*; *Kuo et al., 2017*). This coincides with data from the transgenic mouse model expressing polyQ-expanded human ATXN1 (ATXN1-82Q) in PCs under the control of the *Pcp2* promoter (*Burright et al., 1995*). These mice exhibit aberrant climbing fiber pruning on PC soma and dendrites, a reduction in the responsiveness of PCs to climbing fibers, and alterations in climbing fiber arborization and extension (*Barnes et al., 2011*; *Duvick et al., 2010*; *Ebner et al., 2013*). Previous work has indicated that localization of ATXN1-82Q to the PC nucleus is associated with alterations in climbing fiber-PC synapses (*Ebner et al., 2013*). However, this does not exclude the possibility that affected inferior olive neurons may also influence the degree of climbing fiber synapses on PCs. Other neurodegenerative disorders, including SCA7, 14, and 23, as well as Essential Tremor, have aberrant climbing fiber innervation on PC soma and dendrites, indicating that this observation is not isolated to SCA1 (*Smeets and Verbeek, 2016*). Though the inferior olive is affected in SCA1, the precise role of the inferior olive in the manifestation of SCA1 clinical phenotypes is unclear (*Barnes et al., 2011*; *Duvick et al., 2010*; *Ebner et al., 2013*; *Koeppen et al., 2013*; *Kuo et al., 2017*). Furthermore, the molecular basis of inferior olive alterations in SCA1 is unknown.

The purpose of the present study was to investigate molecular and pathological signatures of the inferior olive in SCA1 mouse models, and further identify similarities and differences in the gene expression profile between the inferior olive and cerebellum, two highly affected tissues in SCA1. We used two SCA1 mouse models, knockin and PC-specific transgenic, which express polyQ-expanded ATXN1 in different cell types and tissues, but faithfully recapture many aspects of the behavioral and pathological phenotypes associated with SCA1 (*Burright et al., 1995*; *Watase et al., 2002*). Comparison of transcriptomics data from these two mouse models allowed for the dissection of how polyQ-expanded ATXN1 influences gene expression changes when it is preferentially expressed in specific cell-types and tissues. We found that differentially regulated genes in the inferior olive segregated into several unique biological pathways. Particularly, the defense response appears largely unique to the SCA1 inferior olive across SCA1 mouse models. We also found that there are common, and distinct, enriched biological pathways between the SCA1 cerebellum and inferior olive.

## Results

### Study design towards understanding the molecular basis of pathogenesis underlying tissue susceptibility in SCA1

A key question in the field is whether the molecular mechanisms underlying pathogenesis in SCA1 affected tissues are similar or distinct. This led us to conduct a thorough comparison of transcriptional changes in the inferior olive and cerebellum using two different SCA1 mouse models, knockin (KI) and transgenic (Tg) (*Figure 1*). Two independent time-points were analyzed. The first time-point selected for analysis was 5 weeks of age, which coincides with the earliest time-point of rotarod impairment in both mouse models (*Clark et al., 1997*; *Watase et al., 2002*). The second time-point chosen was 12 weeks of age, to assess early disease progression. The 12 week time-point also coincides with the initiation of significant thinning of the molecular layer in ATXN1-82Q Tg mice, but precedes gross PC loss.

Since no gender-effects have previously been reported in SCA1 mouse models and to minimize potential gender-effects, if any, three male mice from each genotype were initially used for all RNA-seq experiments. Inferior olive was collected using the decussation of the pyramid and pons as a reference, and the anatomical location was verified under a dissection microscope before processing. RNA-seq data analysis was carried out using the Tuxedo pipeline, using TopHat2 v2.1.0 and Cufflinks v2.2.1 (*Kim et al., 2013a*; *Roberts et al., 2011*; *Trapnell et al., 2012*; *Trapnell et al., 2010*). Differentially expressed genes with a false-discovery rate (FDR) p-value < 0.05 were used in all subsequent analyses unless otherwise stated. Enrichment analysis for biological pathways and molecular functions, as well as upstream regulators, was carried out using NIH DAVID and Ingenuity Pathway Analysis (IPA) (*Huang et al., 2009a*; *Huang et al., 2009b*). Clustering of related biological and molecular pathways was visualized using EnrichmentMap v.3.0 in Cytoscape v.3.6.1 (*Merico et al., 2010*).

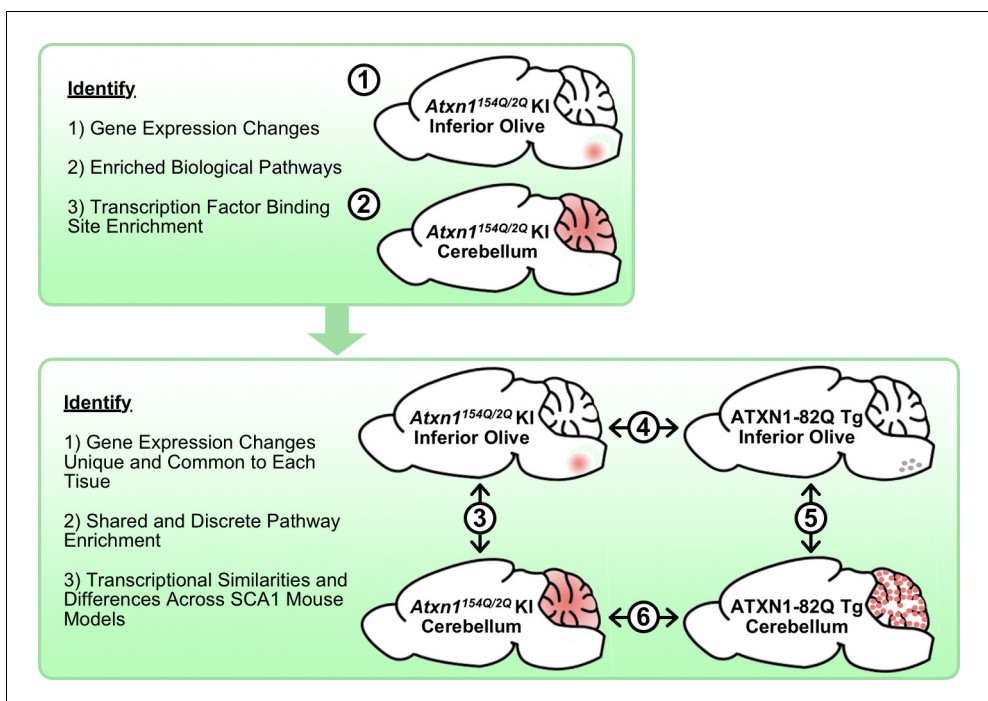

**Figure 1.** Schematic illustrating the cross-tissue and cross-model comparisons conducted to identify common and unique molecular alterations across SCA1 affected tissues. Transcriptomics data from *Atxn1^154Q/2Q* KI inferior olive and cerebellum were first analyzed individually (1-2) before comparing the two tissues (3). Using the ATXN1-82Q Tg inferior olive, comparison of the *Atxn1^154Q/2Q* KI and ATXN1-82Q Tg inferior olive was assessed (4) before evaluating similarities and differences between ATXN1-82Q Tg affected tissues (5). Finally, a cerebellar cross-model comparison was conducted (6).
DOI: https://doi.org/10.7554/eLife.39981.002

We first analyzed the $Atxn1^{154Q/2Q}$ KI mouse model on a pure C57BL/6J background, which expresses polyQ-expanded mutant mouse Atxn1 in appropriate cell types and tissues under the presence of its endogenous regulatory elements (*Watase et al., 2002*) (*Figure 1*). The inferior olive and cerebellum of the $Atxn1^{154Q/2Q}$ KI mouse were first analyzed individually before subsequent temporal and spatial comparisons (*Figure 1*). Using the ATXN1-82Q Tg mouse model on a pure FVB/NJ background, which expresses polyQ-expanded human ATXN1 in PCs due to the *Pcp2* promoter (*Burright et al., 1995*), we assessed similarities and differences in the inferior olive and cerebellum across these two mouse models (*Figure 1*). Using NCBI Protein BLAST, mouse and human ATXN1 (NP_001186234 and NP_001121636, respectively) show 89% identity and 92% positive substitution similarity in amino acid sequence lacking the polyQ tract (*Altschul et al., 1990*).

## SCA1 inferior olive transcriptomics identifies differentially regulated defense response-related genes

We first verified that polyQ-expanded Atxn1 is expressed in the inferior olive of $Atxn1^{154Q/2Q}$ KI mice (*Figure 2A*). A band corresponding to Atxn1-154Q was identified in the inferior olive as well as cerebellar extracts from $Atxn1^{154Q/2Q}$ KI mice (*Figure 2A*). We then conducted RNA-seq in 5 week old $Atxn1^{154Q/2Q}$ KI inferior olive relative to wild-type (WT) control littermates (n = 3 males per genotype) and analyzed the inferior olive differentially regulated genes (*Figure 2—figure supplement 1A*). RNA-seq in the 5 week old $Atxn1^{154Q/2Q}$ KI inferior olive identified 143 annotated transcripts differentially regulated relative to WT control littermates (FDR p-value < 0.05; n = 3 males per genotype) (*Figure 2—figure supplement 1B*). Of these 143 transcripts, 103 were up-regulated (*Figure 2—figure supplement 1B*).

Pathway analysis was carried out using the NIH DAVID Functional Annotation Clustering algorithm (*Huang et al., 2009a*; *Huang et al., 2009b*). Functional Annotation Clustering identifies biological pathways and molecular functions that cluster together based on shared genes, and calculates an enrichment score for each cluster. The top three biological pathway clusters that were significantly enriched in the $Atxn1^{154Q/2Q}$ KI inferior olive at 5 weeks were related to response to organic substance, hormone metabolic process, and neuropeptide hormone activity (*Figure 2—figure supplement 1C*; *Figure 2—source data 1*). The response to organic substance GO term encompasses genes that are altered in response to a change in the activity of a cell or organism due to an organic stimulus. This pathway includes nearly 3000 diverse genes. The response to organic substance pathway in the $Atxn1^{154Q/2Q}$ KI inferior olive included *Ifi27*, which mediates interferon induced apoptosis, and *B2m* and *Ifitm3*, two interferon stimulated genes (ISGs) (*Figure 2—figure supplement 1D*). The Hormone Metabolic Process pathway consisted of a diverse collection of both up- and down-regulated genes, including *Gal*, *Igf2*, and *Wnt4*, representing individual components of galanin, insulin-like growth factor, and Wnt signaling pathways (*Figure 2—figure supplement 1E*).

Given the number of immune-related genes that were segregated into the response to organic substance cluster of pathways, we utilized IPA to determine if any immune-related pathways are found in the $Atxn1^{154Q/2Q}$ KI inferior olive (*Supplementary File 1*). Using a Benjamini-Hochberg corrected p-value, there was significant enrichment for genes associated with virus entry via endocytic pathways in the $Atxn1^{154Q/2Q}$ KI inferior olive relative to WT controls (*Supplementary File 1*). The genes identified in this pathway are largely associated with immune or inflammatory functions. This suggests that a small proportion of immune-related genes are present in the 5 week old $Atxn1^{154Q/2Q}$ KI inferior olive differentially regulated gene list. Candidate upstream regulators were also identified using IPA (*Supplementary File 2*). These transcriptional regulators are identified and ranked based on significantly different genes, and also the gene expression directionality, in the 5 week old $Atxn1^{154Q/2Q}$ KI inferior olive. Estrogen receptor 1 (Esr1) was predicted to be the top upstream regulator of the $Atxn1^{154Q/2Q}$ KI inferior olive transcriptome at 5 weeks of age (*Supplementary File 2*). This coincides with the significant enrichment for hormone metabolic process, neuropeptide hormone activity, and response to corticosteroid pathways identified using NIH DAVID (*Figure 2—figure supplement 1C*).

In the 12 week $Atxn1^{154Q/2Q}$ KI inferior olive, a total of 204 genes were differentially regulated, with 43 annotated transcripts up-regulated and 161 down-regulated (FDR p-value < 0.05; n = 3 males per genotype) (*Figure 2B and C*). Functional annotation clustering identified the defense response, which is the molecular response to a foreign body or injury, as the most highly enriched

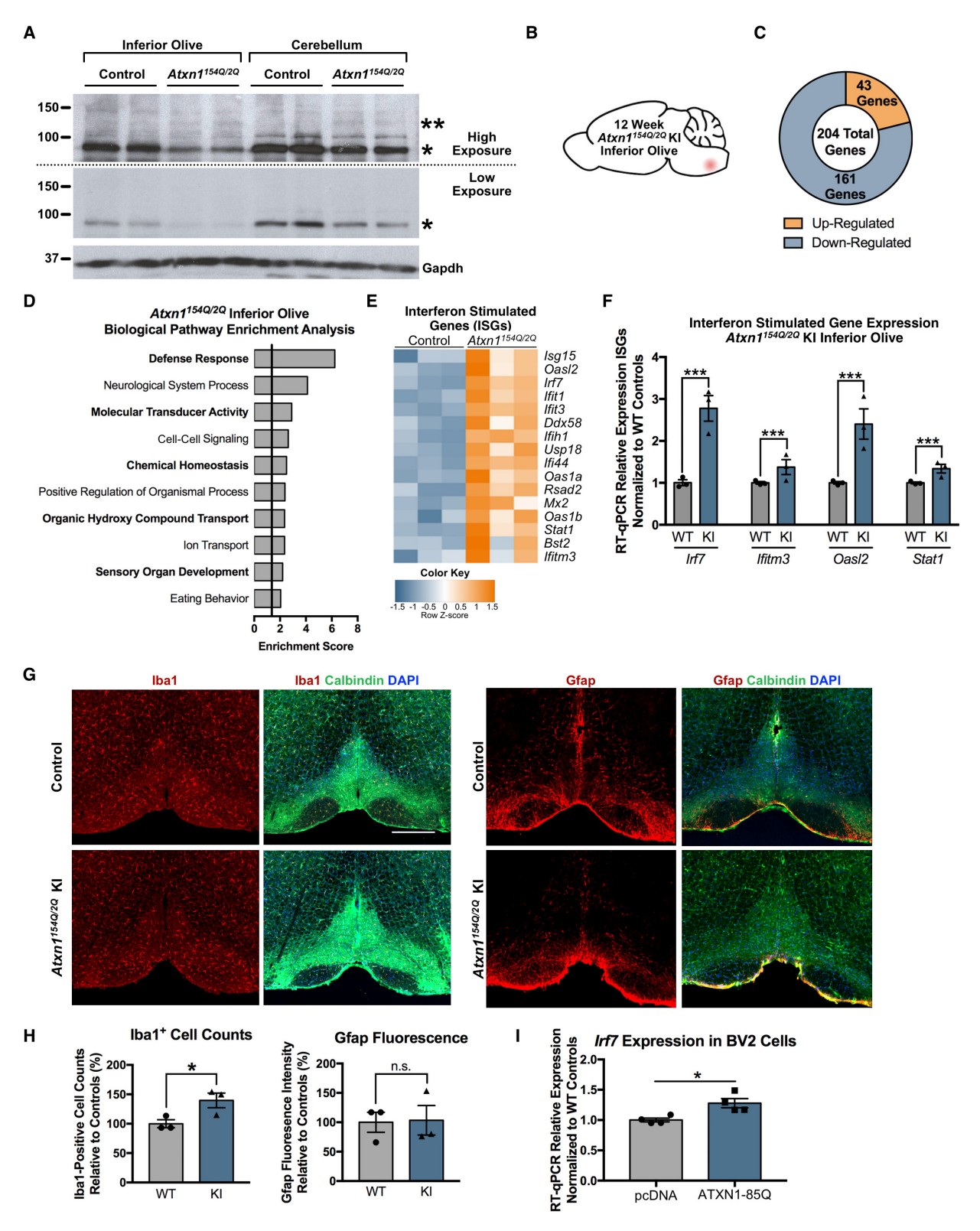

**Figure 2.** Defense Response-related genes are significantly enriched in the 12 week old *Atxn1^154Q/2Q* KI inferior olive. (A) PolyQ-expanded Atxn1 is expressed in the inferior olive of *Atxn1^154Q/2Q* KI mice (n = 2 animals per genotype). * marks Atxn1-2Q and ** marks Atxn1-154Q. (B) Illustration of the brain region examined. (C) Total number of up- and down-regulated genes in the *Atxn1^154Q/2Q* KI inferior olive (FDR p-value < 0.05; n = 3 males per genotype). (D) Biological pathway enrichment for all differentially regulated inferior olive genes. X-axis marks enrichment score, with the significance

*Figure 2 continued on next page*

*Figure 2 continued*

cut-off marked by the vertical black line (p-value < 0.05), in this and all following graphs unless mentioned otherwise. (E) Interferon Stimulated Genes (ISGs) are significantly up-regulated in *Atxn1^{154Q/2Q}* KI. Orange coloring marks up-regulated genes. (F) Validation of up-regulated ISGs in 12 week old *Atxn1^{154Q/2Q}* KI inferior olive using RT-qPCR. All samples normalized to *Gapdh* and *Actb* reference genes. n = 3 males per group. *** p-value < 0.001. Error bars indicate SEM in this and all following graphs. (G) Immunofluorescence staining for Iba1-positive cell counts and Gfap fluorescence intensity imaged in the 12 week old *Atxn1^{154Q/2Q}* KI and WT control inferior olives. Scale bar is 300 μm. (H) Iba1 cell body counts and Gfap fluorescence intensity quantified relative to WT controls as a percentage. *p < 0.05; *t*-test; n.s. = non significant (n = 3 males and females per genotype). (I) *Irf7* mRNA expression is significantly up-regulated in BV2 cells expressing human ATXN1-82Q (*p < 0.05, *t*-test; n = 4 wells per condition).

DOI: https://doi.org/10.7554/eLife.39981.003

The following source data and figure supplements are available for figure 2:

**Source data 1.** 5 week old *Atxn1^{154Q/2Q}* KI inferior olive functional annotation clustering enrichment analysis.
DOI: https://doi.org/10.7554/eLife.39981.008
**Source data 2.** Functional annotation enrichment analysis for the 12 week old *Atxn1^{154Q/2Q}* KI inferior olive.
DOI: https://doi.org/10.7554/eLife.39981.009
**Source data 3.** Upstream regulators for differentially regulated 12 week old *Atxn1^{154Q/2Q}* KI inferior olive genes.
DOI: https://doi.org/10.7554/eLife.39981.010
**Source data 4.** 5 week old *Atxn1^{154Q/2Q}* KI inferior olive enriched pathways for cross time-point overlap with EnrichmentMap (enrichment FDR p-value < 0.05).
DOI: https://doi.org/10.7554/eLife.39981.011
**Source data 5.** 12 week old *Atxn1^{154Q/2Q}* KI Inferior Olive enriched patwhays for cross-time point overlap with EnrichmentMap.
DOI: https://doi.org/10.7554/eLife.39981.012
**Figure supplement 1.** Response to Organic Substance and Hormone Metabolic Process genes are differentially regulated in the 5 week old *Atxn1^{154Q/2Q}* KI inferior olive.
DOI: https://doi.org/10.7554/eLife.39981.004
**Figure supplement 2.** Features of the Defense Response, including Interferon Signaling and IRF Activation, are predicted to have increased activity in the 12 week old *Atxn1^{154Q/2Q}* KI inferior olive.
DOI: https://doi.org/10.7554/eLife.39981.005
**Figure supplement 3.** Subset of genes associated with the enriched biological pathway Neurological System Process are down-regulated in the 12 week old *Atxn1^{154Q/2Q}* KI inferior olive.
DOI: https://doi.org/10.7554/eLife.39981.006
**Figure supplement 4.** Divergent gene expression changes and minimal biological pathway overlap in the temporal comparison of the *Atxn1^{154Q/2Q}* KI inferior olive.
DOI: https://doi.org/10.7554/eLife.39981.007

pathway in the 12 week old inferior olive (*Figure 2D*; *Figure 2—source data 2*). Other pathways associated with the defense response included the immune response and defense response to virus (*Figure 2—source data 2*). These findings are consistent with IPA results, which predicted activation of interferon signaling, activation of IRF by cytosolic pattern recognition receptors, and RIG-1 like receptors in antiviral innate immunity (*Figure 2—figure supplement 2A*). The defense response pathway was composed of a large subset of ISGs, which were largely up-regulated (*Figure 2E*), and were validated using RT-qPCR (*Figure 2F*) (*Irf7* p < 0.0001; *Ifitm3* p = 0.0033; *Oasl2* p < 0.0001; *Stat1* p = 0.0003; *t*-test; n = 3 males per genotype). IPA prediction for upstream regulators influencing gene expression changes identified Irf7 as the transcriptional regulator with the highest enrichment of downstream targets (*Figure 2—figure supplement 2B*; *Figure 2—source data 3*). This was confirmed after inputting inferior olive differentially regulated gene promoters into the MEME-suite FIMO application and assessing significant enrichment for Irf7 binding site DNA motifs relative to shuffled promoter sequences (*Table 1*) (*Bailey et al., 2009*; *Grant et al., 2011*).

Since the defense response was the most significantly enriched biological pathway at the 12 week time-point, and was composed of genes primarily expressed in glial cells, we utilized immunofluorescence staining analyses in 12-week-old *Atxn1^{154Q/2Q}* KI mice to determine if classical markers of astrogliosis and microgliosis, Gfap and Iba1, respectively, were altered in the inferior olive (*Figure 2G*). Relative to WT control animals, there was a significant increase in the number of Iba1-positive cell soma in *Atxn1^{154Q/2Q}* KI inferior olive (n = 3 males and females per genotype; p = 0.0488; *t*-test), but no significant change in Gfap fluorescence intensity at this time-point (n = 3 males and females per genotype; p > 0.05; *t*-test) (*Figure 2G and H*). Since *Irf7* is expressed highly in microglia (*Zhang et al., 2014*), and is the top predicted upstream regulator of inferior olive

**Table 1.** Enrichment for Irf7 binding site DNA motifs in up-regulated 12 week old $Atxn1^{154Q/2Q}$ inferior olive genes.

| Brain region | Motif occurrence | Down-Regulated | Shuffled | p-value | q-value |
|---|---|---|---|---|---|
| Inferior Olive | Present | 32 | 26 | 0.468 | 0.123 |
| | Absent | 128 | 134 | | |
| Brain Region | Motif Occurrence | Up-Regulated | Shuffled | p-value | q-value |
| Inferior Olive | Present | 27 | 2 | 8.8e-9 | 9.2e-9 |
| | Absent | 17 | 42 | | |

FDR p-value calculated using the two-stage linear step-up procedure of Benjamini, Krieger, and Yekutieli (Q 5%).
DOI: https://doi.org/10.7554/eLife.39981.013

differentially regulated genes at the 12 week time-point, (*Figure 2—figure supplement 2B*; *Figure 2—source data 3*), we assessed if *Irf7* expression was altered specifically in microglia in a SCA1 context (*Figure 2I*). The murine microglial BV2 cell line expressed significantly elevated levels of *Irf7* mRNA following overexpression of human ATXN1-82Q (n = 4 wells per experimental group; p = 0.0145; *t-test*) (*Figure 2I*).

Neurological system process was the second most enriched pathway in the 12 week old $Atxn1^{154Q/2Q}$ KI inferior olive, and included genes classically associated with behavior and learning (*Figure 2D*). Some genes from this pathway, including *Calb1, Cck,* and *Kncj10* were down-regulated in the inferior olive (*Figure 2—figure supplement 3A*), and *Calb1* was validated using RT-qPCR (*Figure 2—figure supplement 3B*) (p < 0.01; *t-test*; n = 3 males per genotype). These genes have previously been studied in ataxia research in the cerebellum, suggesting that a subset of ataxia-related genes are also significantly altered in the affected inferior olive of $Atxn1^{154Q/2Q}$ KI mice at the 12 week time-point (*Barski et al., 2003*; *Djukic et al., 2007*; *Ingram et al., 2016*; *Sala-Rabanal et al., 2010*). Collectively, these results indicate that defense response-related genes and neurological system process-related genes are a major feature of the inferior olive transcriptome at the 12 week time-point. Altered expression of defense response-related genes are also found in the 5 week time-point, but they do not represent a significant component.

## Dysregulation of gene expression directionality across two time-points in the $Atxn1^{154Q/2Q}$ KI inferior olive

Identifying similar gene expression and pathway enrichment across the 5 and 12 week time-points in the $Atxn1^{154Q/2Q}$ KI inferior olive may identify commonly conserved molecular mechanisms that contribute to disease onset and early progression (*Figure 2—figure supplement 4A*). We first assessed differential gene expression changes in the 5 and 12 week $Atxn1^{154Q/2Q}$ KI inferior olive relative to their appropriate WT controls (*Figure 2—figure supplement 4B*). Sixty-two annotated transcripts were altered in both the $Atxn1^{154Q/2Q}$ KI 5 week and 12 week inferior olive relative to their WT controls, making up approximately 43.4% of the 5 week differentially regulated gene list and 30.4% of the 12 week differentially regulated gene list (*Figure 2—figure supplement 4B*). Among the 62 genes, 29 genes were consistently up- or down-regulated across time-points (*Figure 2—figure supplement 4B and C*). The log fold change of these genes in the $Atxn1^{154Q/2Q}$ KI relative to WT controls remained stable across both time-points, with no obvious difference in their fold change between the 5 and 12 week time-points (slope = 1.03 ± 0.07; $R^2$ = 0.90) (*Figure 2—figure supplement 4C*). The remaining 33 genes were dysregulated between the two time-points (*Figure 2—figure supplement 4B and C*). Those 33 genes were significantly up-regulated relative to WT controls at 5 weeks, then down-regulated relative to WT controls at 12 weeks of age (*Figure 2—figure supplement 4C*).

To determine what types of biological pathways were conserved across time-points, and which were specific to individual temporal windows, the overlap of biological and molecular pathways in the 5 and 12 week $Atxn1^{154Q/2Q}$ KI inferior olive was assessed (*Figure 2—figure supplement 4D*). Since functional annotation clustering can consolidate related biological pathways and molecular function terms into one broad category, biological themes common to both time-points may be missed. To more easily visualize biological pathways and molecular functions both common, and

unique, to the *Atxn1^154Q/2Q* KI inferior olive across the two time-points, we input all significantly enriched biological and molecular function gene ontology (GO) terms from each time-point into EnrichmentMap (FDR p-value < 0.05) (*Merico et al., 2010*). A key facet of EnrichmentMap is the visualization of multiple GO lists at the same time, which allows for the easy identification of common and divergent biological features across gene expression datasets. Further, consolidating redundant parent and child GO terms can remove noise and allow for easier interpretation of general functional themes in a dataset (*Merico et al., 2010*).

This methodology found enrichment for defense response, receptor activity, and chemical homeostasis-related pathways restricted to the 12 week *Atxn1^154Q/2Q* KI time-point (*Figure 2—figure supplement 4D*; *Figure 2—source data 4* and *5*). There was pathway overlap between the 5 week and 12 week datasets for GO terms related to neuropeptide hormone levels, response to organic substance, and system process (*Figure 2—figure supplement 4D*; *Figure 2—source data 4* and *5*). The majority of the overlapping neuropeptide hormone activity-related genes were down-regulated at both time-points relative to appropriate WT controls, and included the genes *Cck*, *Crh*, *Gal*, and *Nppc* (*Figure 2—figure supplement 4E*; *Figure 2—source data 4* and *5*). The overlapping genes making up the response to organic substance pathway consisted of commonly up-regulated, down-regulated, and dysregulated genes, and included a subset of up-regulated genes linked to the defense response at the 12 week time-point, such as *B2m*, *Iigp1*, *Ifitm3*, and *Usp18* (*Figure 2—figure supplement 4F*; *Figure 2—source data 4* and *5*). The system process pathway was composed of commonly down-regulated and dysregulated genes, such as *Aqp1*, *Rbp1*, and *Ppp1r1b* (*Figure 2—figure supplement 4G*; *Figure 2—source data 4* and *5*). Enrichment for ion transport-related pathways had been identified in the 5 week and 12 week RNA-seq datasets (*Figure 2D*; *Figure 2—figure supplement 1C*), however, the GO term ion transport itself did not reach significance in the 5 week dataset in this analysis (FDR p-value = 0.084).

Collectively, this indicates that a common subset of genes are significantly altered at both the 5 and 12 week time-points in the *Atxn1^154Q/2Q* KI inferior olive. Though the defense response is not significantly enriched at the 5 week time-point, several genes that have functions in the defense response are significantly up-regulated at the 5 week time-point. Further, more than one-half of the commonly altered genes across this temporal window switch directionality, with up-regulation of these genes at the 5 week time-point changing to down-regulation at the 12 week time-point.

## SCA1 cerebellum has preferential enrichment for genes associated with synaptic transmission

We similarly conducted a RNA-seq study in the *Atxn1^154Q/2Q* KI cerebellum at 5 and 12 weeks of age (n = 3 males per genotype) (*Figure 3A*; *Figure 3—figure supplement 1A*). Microarrays have previously been used to assess transcriptional alterations in the *Atxn1^154Q/2Q* KI cerebellum, however, to our knowledge, no RNA-seq experiments have been conducted (*Crespo-Barreto et al., 2010*; *Gatchel et al., 2008*). To allow for subsequent comparison between the inferior olive and cerebellum transcriptome in our analysis, we analyzed the *Atxn1^154Q/2Q* KI cerebellum at appropriate time-points. For the 5 week analysis, only 25 annotated transcripts were differentially regulated compared to WT controls (FDR p-value < 0.05; n = 3 males per genotype) (*Figure 3—figure supplement 1B*). The top functional annotation clustering pathways were associated with reproductive process, transcription factor binding, and cell development pathways, though only reproductive process was significantly enriched based on our enrichment threshold (*Figure 3—figure supplement 1C*; *Figure 3—source data 1*). The genes linked to reproductive process were a diverse subset of genes with differing functions, and included the phosphatase *Dusp1*, the IGF binding protein *Igfbp5*, and the Wnt ligand *Wnt4*, all of which were down-regulated (*Figure 3—figure supplement 1D*). Among the transcription factor binding pathway were the transcriptional activator *Eomes*, as well as *Mdfi*, which is known to regulate Wnt signaling, and the transcriptional repressor *Per1* (*Figure 3—figure supplement 1E*).

In the 12-week-old *Atxn1^154Q/2Q* KI mice cerebella (*Figure 3A*), a total of 943 genes were differentially expressed, with 214 genes up-regulated and 729 genes down-regulated (FDR p-value < 0.05; n = 3 males per genotype) (*Figure 3B*). Pathway enrichment for the entire cerebellar differentially expressed gene list revealed enrichment for pathways associated with chemical synaptic transmission, transmembrane receptor activity, and gated channel activity (*Figure 3C*; *Figure 3—source data 2*). Since the top GO term of chemical synaptic transmission was composed of 90 genes, we

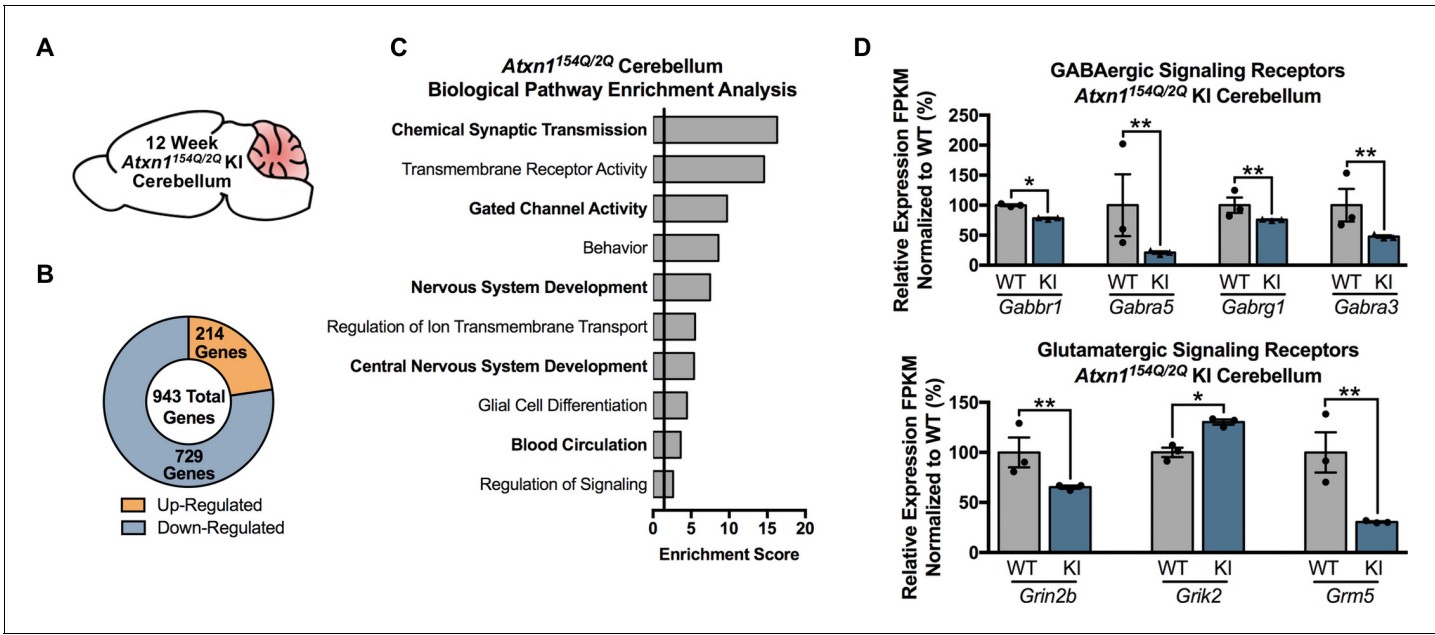

**Figure 3.** Chemical Synaptic Transmission genes, including GABAergic and glutamatergic genes, are differentially regulated in the 12 week old *Atxn1*[154Q/2Q] KI cerebellum. (**A**) Illustration of the tissue examined. (**B**) Total number of up- and down-regulated genes in the 12 week old *Atxn1*[154Q/2Q] KI cerebellum (FDR p-value < 0.05; n = 3 males per genotype for RNA-seq). (**C**) Biological pathway enrichment for all differentially regulated cerebellar genes. (**D**) GABAergic and glutamatergic receptors and receptor subunits are altered in the cerebellum (* FDR p-value < 0.05; ** FDR p-value < 0.01; n = 3 males per group).

DOI: https://doi.org/10.7554/eLife.39981.014

The following source data and figure supplements are available for figure 3:

**Source data 1.** 5 week old *Atxn1*[154Q/2Q] KI cerebellum functional annotation clustering (differentially expressed genes FDR p-value < 0.05).
DOI: https://doi.org/10.7554/eLife.39981.019
**Source data 2.** 12 week old *Atxn1*[154Q/2Q] KI cerebellum functional annotation enrichment analysis.
DOI: https://doi.org/10.7554/eLife.39981.020
**Source data 3.** 5 week old *Atxn1*[154Q/2Q] KI cerebellum functional annotation enrichment analysis (using nominal p-value < 0.01 for genes).
DOI: https://doi.org/10.7554/eLife.39981.021
**Source data 4.** *Atxn1*[154Q/2Q] KI Cerebellum 5 weeks enrichment chart edited for EnrichmentMap (nominal p-value < 0.01).
DOI: https://doi.org/10.7554/eLife.39981.022
**Source data 5.** 12 week old *Atxn1*[154Q/2Q] KI cerebellum enrichment chart edited for EnrichmentMap (FDR p-value < 0.05).
DOI: https://doi.org/10.7554/eLife.39981.023
**Figure supplement 1.** Few genes and pathways are significantly enriched in the 5 week *Atxn1*[154Q/2Q] KI cerebellum.
DOI: https://doi.org/10.7554/eLife.39981.015
**Figure supplement 2.** Consistent gene expression changes in the *Atxn1*[154Q/2Q] KI cerebellum across the 5 and 12 week time-points.
DOI: https://doi.org/10.7554/eLife.39981.016
**Figure supplement 3.** Nominally significant differentially regulated genes in the 5 week old *Atxn1*[154Q/2Q] KI cerebellum overlapped with previously published microarray datasets and revealed significantly enriched pathways.
DOI: https://doi.org/10.7554/eLife.39981.017
**Figure supplement 4.** Consistent gene expression changes and pathway enrichment in the 5 week and 12 week *Atxn1*[154Q/2Q] KI cerebellum with an expanded 5 week p-value cutoff.
DOI: https://doi.org/10.7554/eLife.39981.018

assessed the molecular function of those genes in greater detail. The most enriched molecular function within this grouping of genes was neurotransmitter receptor activity, and included genes linked to GABAergic and glutamatergic signaling, as well as serotonin and acetylcholine signaling (*Figure 3D*; *Figure 3—source data 2*). The remaining genes were largely associated with voltage-gated calcium channel activity and calmodulin binding (*Figure 3—source data 2*).

Upstream regulator prediction for the 12 week *Atxn1*[154Q/2Q] KI cerebellum transcriptomics dataset revealed Creb1 as one of the top candidates for regulating downstream targets

(*Supplementary File 3*). However, previous studies have identified the transcriptional repressor Capicua (Cic) as an Atxn1 interactor, and a growing body of evidence suggests that the Atxn1-Cic interaction may be important in driving molecular alterations and pathological and behavioral phenotypes in the SCA1 cerebellum (*Fryer et al., 2011*; *Ingram et al., 2016*; *Kim et al., 2013b*; *Lam et al., 2006*; *Lim et al., 2006*; *Rousseaux et al., 2018*). Since Cic motifs are not found in IPA, we queried whether Cic binding site DNA motifs were also significantly enriched in the promoter region of 12 week $Atxn1^{154Q/2Q}$ KI cerebellum differentially regulated genes (*Table 2*). Relative to shuffled nucleotide sequences, there was strong enrichment for Cic binding site DNA motifs in the 12 week $Atxn1^{154Q/2Q}$ KI cerebellum among down-regulated genes (*Table 2*). To determine if Cic binding site motifs were only present in the promoter region of cerebellum differentially regulated genes, the inferior olive was also tested (*Table 2*). There was enrichment among up-regulated inferior olive genes at 12 weeks of age for Cic binding sites, but the degree of enrichment was not as substantial as in the cerebellum (*Table 2*).

## Conserved gene expression changes and pathway enrichment in the $Atxn1^{154Q/2Q}$ KI cerebellum across disease progression time-points

Despite the few genes altered in the 5 week $Atxn1^{154Q/2Q}$ KI cerebellum, we next assessed whether those genes overlapped with the 12 week cerebellum differentially regulated gene list (*Figure 3—figure supplement 2A*). Of the 25 genes differentially regulated in the 5 week old $Atxn1^{154Q/2Q}$ KI cerebellum, 17 overlapped with the 12 week cerebellum dataset (*Figure 3—figure supplement 2B*). The majority of these were commonly down-regulated (*Figure 3—figure supplement 2B and C*). In fact, all of the genes significantly altered in both the $Atxn1^{154Q/2Q}$ KI 5 week and 12 week cerebellum relative to appropriate WT controls were differentially regulated in the same direction (*Figure 3—figure supplement 2B and C*). Analysis of these genes found a larger fold change in 12 week $Atxn1^{154Q/2Q}$ KI relative to controls than in the 5 week dataset (slope = 1.3 ± 0.13, $R^2$ = 0.86), indicating a slight progression in the severity of the up- or down-regulation of genes across the temporal window (*Figure 3—figure supplement 2C*). However, this analysis was conducted with only 17 genes, and the interpretation may change with a larger gene set. The commonly down-regulated

**Table 2.** Enrichment for Cic binding site DNA motifs in down- and up-regulated 12 week old $Atxn1^{154Q/2Q}$ inferior olive and cerebellum.

| Cic DNA motif | Brain region | Motif occurrence | Down-Regulated | Shuffled | p-value | q-value |
|---|---|---|---|---|---|---|
| TGAATGAA | Inferior Olive | Present | 31 | 19 | 0.090 | 0.095 |
| | | Absent | 129 | 141 | | |
| | Cerebellum | Present | 126 | 57 | 5.6e-8 | 2.9e-7 |
| | | Absent | 604 | 673 | | |
| TGAATGGA | Inferior Olive | Present | 35 | 25 | 0.197 | 0.148 |
| | | Absent | 125 | 135 | | |
| | Cerebellum | Present | 157 | 104 | 3.7e-4 | 9.7e-4 |
| | | Absent | 573 | 626 | | |
| Cic DNA Motif | Brain Region | Motif Occurrence | Up-Regulated | Shuffled | p-value | q-value |
| TGAATGAA | Inferior Olive | Present | 10 | 4 | 0.143 | 0.125 |
| | | Absent | 34 | 40 | | |
| | Cerebellum | Present | 32 | 19 | 0.073 | 0.095 |
| | | Absent | 181 | 194 | | |
| TGAATGGA | Inferior Olive | Present | 13 | 2 | 0.003 | 0.005 |
| | | Absent | 31 | 42 | | |
| | Cerebellum | Present | 38 | 29 | 0.287 | 0.188 |
| | | Absent | 175 | 184 | | |

FDR p-value calculated using the two-stage linear step-up procedure of Benjamini, Krieger, and Yekutieli (Q 5%).
DOI: https://doi.org/10.7554/eLife.39981.024

genes, which made up the majority of the overlapping gene set, included the transcription factors *Eomes* and *Mdfi*, as well as IGF binding protein *Igfbp5* (**Figure 3—figure supplement 2D**). Among the three up-regulated genes shared between the 5 week and 12 week datasets were *Ifi203*, which responds to interferon-beta, *Necab*1, a calcium binding protein, and the synaptic vesicle protein *Synpr* (**Figure 3—figure supplement 2E**).

We were unable to assess enrichment for similar biological pathways between the 5 week and 12 week time-points, likely due to the small number of genes altered at 5 weeks in the *Atxn1^{154Q/2Q}* KI cerebellum. The 5 week *Atxn1^{154Q/2Q}* KI dataset may have yielded few significantly different genes due to animal variability, small sample size (n = 3 males per genotype), or the stringent FDR p-value < 0.05 cutoff (nominal p-value = 5.0E-05). To allow for further temporal and cross-tissue comparisons, we sought to extend our p-value cutoff for the 5 week old *Atxn1^{154Q/2Q}* KI cerebellum. Previous microarrays in the *Atxn1^{154Q/2Q}* KI cerebellum at 4 weeks and 7 weeks of age found a large number of significantly different genes relative to WT controls (**Crespo-Barreto et al., 2010**; **Gatchel et al., 2008**). We assessed our 5 week dataset in comparison with the previously published 4 week old cohort, and found a portion of the nominally significant genes (nominal p-value < 0.01) that overlapped with the previously published microarray (**Figure 3—figure supplement 3A**). The directionality of the gene expression changes were largely consistent (**Figure 3—figure supplement 3A**). Due to the overlap between the two datasets, we extended our p-value for the 5 week old *Atxn1^{154Q/2Q}* KI cerebellum to include genes with a nominal p-value < 0.01. With this analysis, 128 genes were significantly different relative to WT controls (**Figure 3—figure supplement 3B**). The majority of genes, approximately 72%, were down-regulated (**Figure 3—figure supplement 3B**). Pathway analysis with these genes revealed significant enrichment for pathways related to regulation of cell communication, signaling receptor activity, and chemical synaptic transmission, which was similar to those found in the 12 week old *Atxn1^{154Q/2Q}* KI cerebellum (**Figure 3C**; **Figure 3—figure supplement 3C**; **Figure 3—source data 3**).

The comparison between the 5 week and 12 week *Atxn1^{154Q/2Q}* KI cerebellum relative to appropriate WT controls using the nominal p-value < 0.01 for the 5 week old group revealed a larger degree of overlap between the two temporal time-points (**Figure 3—figure supplement 4A, B**). Of the 76 genes that were differentially regulated at both time-points relative to their WT littermates, 73 genes were altered in the same direction (**Figure 3—figure supplement 4B, C**). The genes commonly up- and down-regulated at both time-points trended toward greater log fold changes at the 12 week time-point than the 5 week (Slope = 1.53 ± 0.10; $R^2$ = 0.78) (**Figure 3—figure supplement 4C′**). This indicates that genes significantly up- or down-regulated at 12 weeks had a larger fold change than when they were dysregulated at 5 weeks of age (**Figure 3—figure supplement 4C′**).

To determine what pathways were similar, and unique, to each time-point, pathway enrichment was assessed among the *Atxn1^{154Q/2Q}* KI cerebellum at 5 weeks (nominal p-value < 0.01) and 12 weeks (FDR p-value < 0.05) (**Figure 3—figure supplement 4D**; **Figure 3—source data 4** and **5**). There was overlap between the 5 and 12 week time-points for pathways related to synaptic signaling, which could be further broken down into smaller clusters of cell-cell signaling, gated channel activity, and system process (**Figure 3—figure supplement 4D**; **Figure 3—source data 4** and **5**). There was also pathway overlap in CNS development (**Figure 3—figure supplement 4D**; **Figure 3—source data 4** and **5**). Among the commonly enriched pathways, the majority of the up-regulated genes were associated with the CNS development pathway (**Figure 3—figure supplement 4E**). Commonly down-regulated genes made up a substantial portion of the cell-cell signaling and gated channel activity pathways, which were subcomponents of the synaptic signaling cluster (**Figure 3—figure supplement 4F**). This suggests that biological and molecular pathways identified at 12 weeks in the *Atxn1^{154Q/2Q}* KI cerebellum are already present at 5 weeks, though to a lesser extent. Further, the directionality of the gene expression changes at 5 weeks is largely conserved at 12 weeks in the *Atxn1^{154Q/2Q}* KI cerebellum.

## Comparison between cerebellum and inferior olive reveals largely divergent pathway enrichment

Next, we assessed whether molecular alterations found in the inferior olive and cerebellum of *Atxn1^{154Q/2Q}* KI mice are similar or distinct. To analyze the similarities between these two affected tissues in greater detail, the overlap in the two SCA1 affected tissues was assessed at each time-point (**Figure 4**; **Figure 4—figure supplement 1**). In the 5 week old *Atxn1^{154Q/2Q}* KI inferior olive

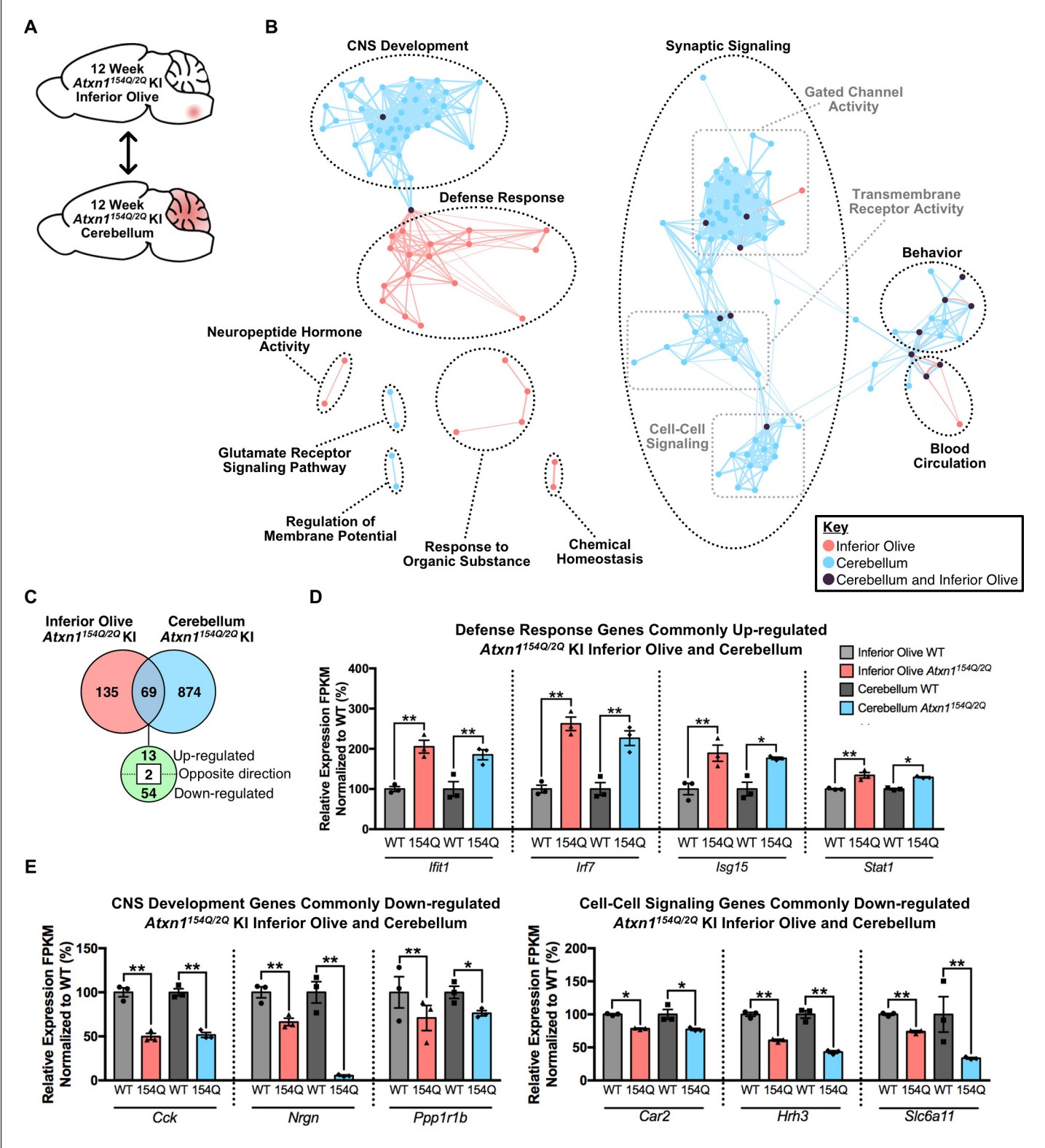

**Figure 4.** Cross-tissue comparison of pathway enrichment reveals common and unique features of the 12 week old *Atxn1^{154Q/2Q}* KI inferior olive and cerebellum. (A) Schematic of the cross-tissue comparison conducted. (B) Clustering of GO terms from the 12 week old inferior olive and cerebellum differentially expressed genes lists. In this and all following figures, nodes represent GO terms and edges connect nodes that share common genes. Edge width corresponds to the number of genes shared between nodes, and edge length represents the similarity coefficient between nodes. Color-coded nodes represent GO terms from either the inferior olive (pink), cerebellum (blue), or GO terms shared by both tissues (purple). All GO terms

*Figure 4 continued on next page*

*Figure 4 continued*

were significantly enriched within the datasets (FDR p-value < 0.05). (C) Total number of differentially regulated genes that are common, and uniquely altered, in the 12 week old *Atxn1*$^{154Q/2Q}$ KI inferior olive and cerebellum (FDR p-value < 0.05). (D) A subset of genes commonly up-regulated and (E) commonly down-regulated in both the inferior olive and cerebellum relative to WT controls (* FDR p-value < 0.05; ** FDR p-value < 0.01; n = 3 males per genotype for RNA-seq).

DOI: https://doi.org/10.7554/eLife.39981.025

The following source data and figure supplement are available for figure 4:

**Source data 1.** 5 week old *Atxn1*$^{154Q/2Q}$ KI Inferior Olive enrichment chart for EnrichmentMap (FDR p < 0.05).
DOI: https://doi.org/10.7554/eLife.39981.027
**Source data 2.** 5 week old *Atxn1*$^{154Q/2Q}$ KI cerebellum enrichment chart edited for EnrichmentMap (nominal p-value < 0.01).
DOI: https://doi.org/10.7554/eLife.39981.028
**Source data 3.** 12 week old *Atxn1*$^{154Q/2Q}$ KI inferior olive enrichment chart for EnrichmentMap.
DOI: https://doi.org/10.7554/eLife.39981.029
**Source data 4.** 12 week old *Atxn1*$^{154Q/2Q}$ KI cerebellum enrichment chart for EnrichmentMap.
DOI: https://doi.org/10.7554/eLife.39981.030
**Figure supplement 1.** Minimal gene overlap in the 5 week old *Atxn1*$^{154Q/2Q}$ KI cerebellum and inferior olive in the cross-tissue comparison.
DOI: https://doi.org/10.7554/eLife.39981.026

and cerebellum comparison, 14 genes overlapped, 9 of which were regulated in opposing directions in each tissue (*Figure 4—figure supplement 1A and B*). Biological and molecular pathway overlap consisted of a single node in the response to organic substance cluster (*Figure 4—figure supplement 1C*; *Figure 4—source data 1* and *2*). This pathway consisted of a collection of the up-regulated, down-regulated, and dysregulated genes that were altered in both the inferior olive and cerebellum (*Figure 4—figure supplement 1D*). These results indicate that minimal pathway and gene expression overlap exists in the 5 week old *Atxn1*$^{154Q/2Q}$ KI inferior olive and cerebellum, and that gene expression changes in these two tissues at this time-point are brain-region specific.

In the 12 week *Atxn1*$^{154Q/2Q}$ KI inferior olive and cerebellum comparison, GO annotations segregated into biological and molecular functions that were also largely distinct depending on tissue (*Figure 4A and B*; *Figure 4—source data 3* and *4*). Node clusters that appeared unique to the inferior olive were composed of pathways associated with the defense response, response to organic substance, neuropeptide hormone activity, and chemical homeostasis (*Figure 4B*; *Figure 4—source data 3*). In contrast, node clusters made up entirely of differentially expressed genes originating from the cerebellum included regulation of membrane potential and glutamate receptor signaling pathway (*Figure 4B*; *Figure 4—source data 4*). Four node clusters contained a mixture of GO terms enriched in both the inferior olive and cerebellum, including CNS development, blood circulation, behavior, and synaptic signaling (*Figure 4B*, *Figure 4—source data 3* and *4*). The synaptic signaling cluster was defined as a large cluster based on our clustering criteria with three minor groups segregating within that cluster (*Figure 4B*). Those three groups were gated chanel activity, transmembrane receptor activity, and cell-cell signaling (*Figure 4B*). There was a higher degree of overlap between the inferior olive and cerebellum GO terms enriched within these clusters, indicating a degree of tissue homogeneity. However, the cell-cell signaling cluster only had one node that overlapped between tissues (*Figure 4B*, *Figure 4—source data 3* and *4*). While this suggests a degree of overlap between the two tissues in cell-cell signaling, it also indicates that this biological pathway is not a major feature of the inferior olive transcriptome.

In total, 69 common genes were differentially regulated in both tissues relative to appropriate WT controls, with 13 genes commonly up-regulated and 54 down-regulated (*Figure 4C*). Of the commonly up-regulated genes, four were ISGs (*Figure 4D*). Among the shared down-regulated genes were those associated with CNS development and cell-cell signaling, collectively indicating that there is a subset of shared genes and pathways between the two tissues that may mediate pathogenesis (*Figure 4E*).

## Defense response-related pathways are enriched in the inferior olive across SCA1 mouse models

The defining feature of the 12 week *Atxn1*$^{154Q/2Q}$ KI inferior olive was the defense response, suggesting that the major cell-type undergoing transcriptional alterations in the *Atxn1*$^{154Q/2Q}$ KI inferior

olive may be from a glial origin (*Figure 2D, E, G and H*). In addition, a subset of ISGs up-regulated at 12 weeks were also up-regulated at 5 weeks of age, indicating that alterations in these genes already occurs at an early symptomatic time-point (*Figure 2—figure supplement 1D*). This up-regulation of defense response-related genes may be due to a direct effect of polyQ-expanded Atxn1 in glial cells (*Figure 2I*), or it may occur in response to polyQ-expanded Atxn1 effects in neurons. Consistent with this idea, previous studies have shown an elevation of Gfap and Iba1 proteins in the ATXN1-82Q Tg mouse cerebellum (*Cvetanovic et al., 2015*). Expression of ATXN1-82Q in these Tg mice is driven by the *Pcp2* promoter, thus isolating the expression of polyQ-expanded ATXN1 to PCs (*Burright et al., 1995*). This model would therefore allow for the identification of non-cell autonomous effects in the inferior olive, and potentially determine if the defense response enrichment in the 12 week *Atxn1^{154Q/2Q}* KI inferior olive is due to non-cell autonomous effects of polyQ-expanded Atxn1.

The ATXN1-82Q Tg mice exhibit robust overexpression of *ATXN1-82Q* mRNA in the cerebellum, but show no significant increase relative to WT control littermates in the inferior olive (*Figure 5A*). Protein quantification of ATXN1-82Q found that, while it is expressed in whole cerebellar lysate, it does not resolve in the inferior olive (*Figure 5B*). This verified that ATXN1-82Q Tg mice do not express polyQ-expanded ATXN1 in the inferior olive, which is consistent with a previous report (*Burright et al., 1995*). Further, it indicates that transcriptional alterations identified in the ATXN1-82Q Tg inferior olive, if any, are likely due to non-cell autonomous effects of polyQ-expanded ATXN1 in the cerebellum, and not due to its expression in the inferior olive.

Analysis of RNA-seq data from 5 week old ATXN1-82Q Tg inferior olive identified 148 genes that were differentially regulated relative to their WT control littermates (FDR $p < 0.05$; n = 3 males per genotype) (*Figure 5—figure supplement 1A and B*). The majority of genes (142 out of 148 genes) were up-regulated (*Figure 5—figure supplement 1B*). Pathway analysis found enrichment for vasculature development, enzyme-linked receptor protein signaling pathway, and cellular response to organic substance in the ATXN1-82Q Tg inferior olive at 5 weeks of age (*Figure 5—figure supplement 1C*; *Figure 5—source data 1*). No significant enrichment for defense response-related pathways was identified at this time-point using NIH DAVID or IPA (*Figure 5—source data 1*; *Supplementary file 4*). However, some immune-related genes were segregated into the vasculature development cluster (*Figure 5—figure supplement 1D*). *Lbp* and *Nov* (also known as *Ccn3*) have immune functions, and were significantly up-regulated at 5 weeks in the ATXN1-82Q Tg inferior olive (*Figure 5—figure supplement 1D*). Genes associated with enzyme linked receptor protein signaling pathway were also largely up-regulated, and included *Bmp7, Folr1, Gdf7*, and *Kl* (*Figure 5—figure supplement 1E*).

The 12 week ATXN1-82Q Tg inferior olive RNA-seq data revealed 126 genes differentially regulated (FDR p-value $< 0.05$; n = 3 males per genotype) (*Figure 5—figure supplement 2A and B*). Approximately 51% of the differentially regulated genes were up-regulated (*Figure 5—figure supplement 2B*). These genes represented pathways related to those identified at 5 weeks, and included blood vessel morphogenesis and response to organic substance (*Figure 5—figure supplement 2C*; *Figure 5—source data 2*), which consisted of both up- and down-regulated genes (*Figure 5—figure supplement 2D*). Interestingly, a cluster of pathways including defense response, regulation of defense response, cellular response to interleukin-I, and complement and coagulation cascade were among the top ten pathways enriched in the 12 week inferior olive (*Figure 5—figure supplement 2C*; *Figure 5—source data 2*).

To identify similarities in gene expression and pathway enrichment across the ATXN1-82Q Tg inferior olive, we assessed the 5 and 12 week ATXN1-82Q Tg inferior olive together (*Figure 5—figure supplement 3A*). A total of 61 genes were altered in both the 5 week and 12 week time-points relative to their WT controls (*Figure 5—figure supplement 3B*). However, 51 of these common genes were regulated in opposing directions (*Figure 5—figure supplement 3B*). These genes where up-regulated in the 5 week inferior olive, and down-regulated at the 12 week time-point (*Figure 5—figure supplement 3C*). The 10 genes that were up-regulated at both time-points relative to appropriate WT controls had a minor decrease in fold change over time (Slope = $0.91 \pm 0.12$; $R^2 = 0.87$), indicating the degree of up-regulation remained largely stable at the 5 and 12 week time-points (*Figure 5 — figure supplement 3C'*). Pathway analysis revealed that the defense response was enriched only at the 12 week time-point (*Figure 5—figure supplement 3D*; *Figure 5—source data 3 and 4*). There was a small amount of overlap between the two time-points, including overlap

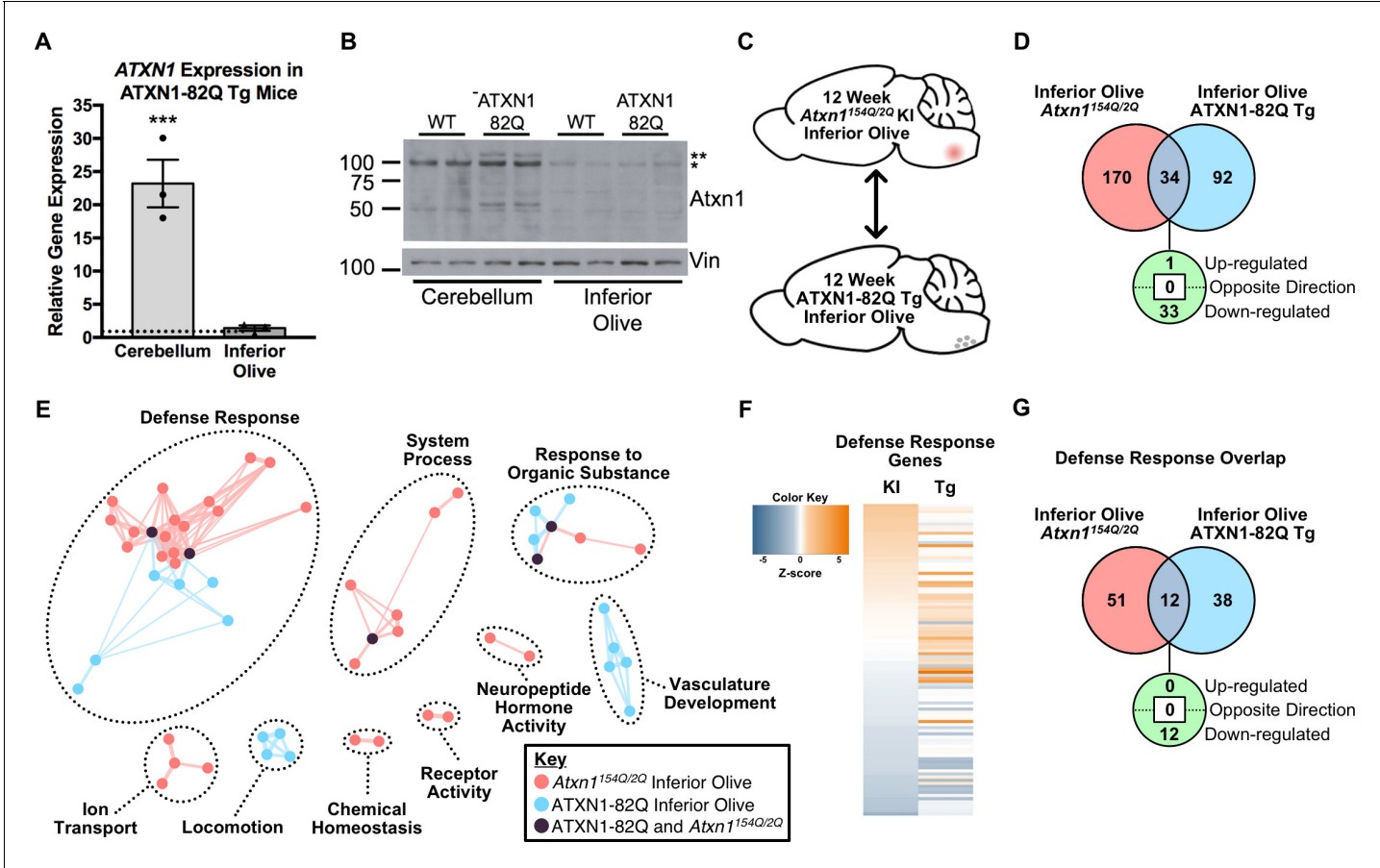

**Figure 5.** The Defense Response is enriched in both the 12 week old *Atxn1*[154Q/2Q] KI inferior olive and ATXN1-82Q Tg inferior olive. (**A**) *ATXN1* mRNA expression in the 12 week old ATXN1-82Q Tg cerebellum and inferior olive (***p < 0.001, *t-test*; n = 3 males per genotype). (**B**) PolyQ-expanded ATXN1 is present in the cerebellum, but not in the inferior olive (n = 2 males per genotype). * marks Atxn1-2Q, ** marks ATXN1-82Q.(**C**) Schematic of the cross-model, 12 week old inferior olive comparison. (**D**) Total number of differentially regulated genes that are common, and uniquely altered, in the inferior olive of *Atxn1*[154Q/2Q] KI and ATXN1-82Q Tg mice (FDR p-value < 0.05). (**E**) Clustering of GO terms from the *Atxn1*[154Q/2Q] KI and ATXN1-82Q Tg inferior olive differentially expressed genes lists. (**F**) Heatmap of log fold changes in Defense Response-related genes relative to control littermates from both the 12 week *Atxn1*[154Q/2Q] KI and 12 week ATXN1-82Q Tg inferior olive. (**G**) Total number of differentially regulated genes linked to Defense Response that are common, and uniquely altered, in the 12 week old *Atxn1*[154Q/2Q] KI and ATXN1-82Q Tg inferior olive (FDR p-value < 0.05).
DOI: https://doi.org/10.7554/eLife.39981.031

The following source data and figure supplements are available for figure 5:

**Source data 1.** 5 week old ATXN1-82Q Tg inferior olive functional annotation clustering analysis.
DOI: https://doi.org/10.7554/eLife.39981.035

**Source data 2.** 12 week old ATXN1-82Q Tg inferior olive functional annotation clustering.
DOI: https://doi.org/10.7554/eLife.39981.036

**Source data 3.** 5 week old ATXN1-82Q Tg inferior olive enrichment chart for EnrichmentMap.
DOI: https://doi.org/10.7554/eLife.39981.037

**Source data 4.** 12 week old ATXN1-82Q Tg inferior olive enrichment chart for EnrichmentMap.
DOI: https://doi.org/10.7554/eLife.39981.038

**Figure supplement 1.** The majority of genes significantly altered in the 5 week ATXN1-82Q Tg inferior olive are up-regulated.
DOI: https://doi.org/10.7554/eLife.39981.032

**Figure supplement 2.** Differentially regulated genes identified in the 12 week ATXN1-82Q Tg inferior olive are associated with Blood Vessel Morphogenesis, Response to Organic Substance, and Defense Response-related pathways.
DOI: https://doi.org/10.7554/eLife.39981.033

**Figure supplement 3.** Divergent gene expression changes and minimal biological pathway overlap in the temporal comparison of the ATXN1-82Q Tg inferior olive.
DOI: https://doi.org/10.7554/eLife.39981.034

between the response to organic substance and vasculature development pathways (**Figure 5—figure supplement 3D**; **Figure 5—source data 3** and **4**). The response to organic substance and vasculature development clusters consisted of some commonly up-regulated and dysregulated genes (**Figure 5—figure supplement 3E and F**; **Figure 5—source data 3** and **4**).

Enrichment for defense response-related genes at the 12 week ATXN1-82Q Tg time-point suggests there may be some similar genetic changes in the ATXN1-82Q Tg and *Atxn1^154Q/2Q* KI inferior olive (**Figure 5C, D and E**). In total, 34 genes were commonly altered in both the ATXN1-82Q Tg and the *Atxn1^154Q/2Q* KI inferior olive relative to their WT controls at 12 weeks of age, with 33 of those genes commonly down-regulated (**Figure 5D**). To identify common and uniquely enriched pathways in these tissues, GO annotations were input into EnrichmentMap and a network was generated (**Figure 5E**). There was clustering between the *Atxn1^154Q/2Q* KI and ATXN1-82Q Tg inferior olive for GO terms related to the defense response, with some overlap in the response to organic substance and system process clusters (**Figure 5E**).

The minimal overlap for defense response-related pathways between the *Atxn1^154Q/2Q* KI and ATXN1-82Q Tg 12 week inferior olive suggests that unique genes may be constituting the defense response in each mouse model. To ascertain whether the same genes were driving enrichment for the defense response in the inferior olive, we compared the genes represented in each of the mouse models at 12 weeks of age (**Figure 5F and G**). An assessment of all defense response-related genes from both mouse models found a marked difference in expression directionality (**Figure 5F**). Genes up-regulated in the ATXN1-82Q Tg are largely non-significant in the *Atxn1^154Q/2Q* KI, and genes up- or down-regulation in the *Atxn1^154Q/2Q* KI are largely non-significant or regulated in opposing directions (**Figure 5F**). A total of 12 defense response-related genes were commonly altered in each of the two groups out of a combined total of 101, with all of them commonly down-regulated (**Figure 5G**). Collectively, this suggests that different transcriptional regulators are driving distinct alterations in defense response-related genes across the SCA1 animal models at 12 weeks of age (**Figure 5F and G**). Indeed, IPA prediction for upstream regulators of the ATXN1-82Q Tg inferior olive transcriptome at 12 weeks identified Esr1 and Tcl1a as the top potential upstream regulators, and no significant enrichment for Irf7 was observed (**Supplementary File 5**).

## Distinct, brain region-specific pathway enrichment is conserved in a transgenic mouse model of SCA1

The over-representation of the defense response in the ATXN1-82Q Tg inferior olive suggests that non-cell autonomous mechanisms may alter astrocyte or microglia reactivity in the SCA1 inferior olive. To determine if enrichment for defense response-related genes is specific to the inferior olive in the ATXN1-82Q Tg mouse model, we conducted RNA-seq in the cerebellum at 5 and 12 weeks of age. Sequencing in the 5 week ATXN1-82Q Tg mouse cerebellum revealed 402 genes significantly different relative to WT control littermates (**Figure 6—figure supplement 1A and B**). Of the 402 genes differentially regulated, 350 were down-regulated (**Figure 6—figure supplement 1B**). The biological and molecular pathways these genes made up were chemical synaptic transmission, regulation of cell communication, and cation transmembrane transport (**Figure 6—figure supplement 1C**; **Figure 6—source data 1**). The chemical synaptic transmission pathway included genes related to GABAergic signaling, glutamatergic signaling, and scaffolding proteins (**Figure 6—figure supplement 1D**; **Figure 6—source data 1**). These were largely down-regulated (**Figure 6—figure supplement 1D**) Among the regulation of cell communication pathway were the genes *Baiap2*, *Kalrn*, and *Clstn2*, which were also down-regulated (**Figure 6—figure supplement 1E**; **Figure 6—source data 1**).

A greater number of genes were significantly altered at the 12 week time-point, with 1062 annotated transcripts significantly altered in the ATXN1-82Q Tg cerebellum, with approximately 43% of genes up-regulated (**Figure 6—figure supplement 2B**). The top biological and molecular functions enriched in this tissue were synaptic signaling, regulation of signaling, and neurological system process (**Figure 6—figure supplement 2C**; **Figure 6—source data 2**). Glutamatergic receptors made up a portion of the synaptic signaling pathway, and consisted of a collection of up- and down-regulated genes (**Figure 6—figure supplement 2D**; **Figure 6—source data 2**). The regulation of signaling pathway contained similar genes as the synaptic signaling pathway, but a major component of the regulation of signaling pathway was Rho GTPases, which were up- and down-regulated (**Figure 6—figure supplement 2D**; **Figure 6—source data 2**).

Some pathways identified in the 5 week and 12 week dataset appeared to be quite similar, including synaptic signaling, chemical synaptic transmission, and ion transport (*Figure 6—figure supplements 1C* and *2C*). To determine the extent of the similarities, we conducted a more thorough comparison between the two time-points (*Figure 6—figure supplement 3A*). Between the two datasets, 323 genes were differentially regulated at both time-points relative to their WT controls (*Figure 6—figure supplement 3B*). Interestingly, the majority of genes (316 out of the 323 genes) were regulated in the same direction (*Figure 6—figure supplement 3C and C'*). Among the 316 genes altered in the same direction over time, there appeared to be an increase in the severity of the log fold change from the 5 week to 12 week time-point (slope = 1.47 ± 0.03, $R^2$ = 0.89) (*Figure 6 - figure supplement 3C*). This suggests that a subset of gene expression changes occurring as early as 5 weeks become progressively more severe with age (*Figure 6 - figure supplement 3C*). An examination of pathway overlap between the two time-points identified pathways related to cell-cell signaling, chemical synaptic transmission/ion transport, and system process (*Figure 6—figure supplement 3D*). Pathways related to CNS development, receptor activity, and movement of cell were specific to the 12 week time-point (*Figure 6—figure supplement 3D*). The cell-cell signaling, chemical synaptic transmission/ion transport, and system process pathways consisted of a collection of both up- and down-regulated genes, such as *Casp3*, *Gfap*, *Calb1*, *Baiap2*, *Kalrn*, and *Gabbr1* (*Figure 6—figure supplement 3E*). Collectively, this indicates that the 5 week ATXN1-82Q Tg cerebellum differentially regulated gene list largely overlaps with the 12 week dataset, though it appears that many changes in gene expression occur after the 5 week time-point.

To determine if brain region-dependent gene expression changes and pathway enrichment were occurring in the ATXN1-82Q Tg mouse, a tissue comparison was conducted at both time-points. At the 5 week time-point (*Figure 6—figure supplement 4A*), 23 annotated transcripts were differentially regulated in both the 5 week inferior olive and cerebellum (*Figure 6—figure supplement 4B*). However, 15 of the 23 genes were altered in opposing directions (*Figure 6—figure supplement 4B*). There was minor overlap in terms of enriched biological and molecular pathways between the two tissues, with only one node shared between the two brain regions in the hormone activity cluster (*Figure 6—figure supplement 4C*). At the 12 week time-point (*Figure 6A*), 59 genes were commonly altered in both tissues (*Figure 6B*). There were 15 up-regulated genes, three of which constitute components of the defense response, specifically the complement cascade (i.e. *C1qa*, *C1qb*, *C1qc*) (*Figure 6C*). However, the majority of the genes commonly altered in both the 12 week ATXN1-82Q Tg cerebellum and inferior olive were altered in opposing directions, some of which were associated with cell growth and development (*Figure 6C*).

Comparison of the enriched GO terms from the ATXN1-82Q Tg inferior olive and cerebellum found many biological pathways that are distinct between the tissues, such as nervous system development, ion transport and cell-cell signaling (*Figure 6D*). The defense response and vasculature development clusters were the only two clusters largely specific to the inferior olive (*Figure 6D*). Collectively, this indicates that defense response-related genes are not a major feature of the cerebellar transcriptome in ATXN1-82Q Tg mice at 12 weeks of age. This data also supports our findings in *Atxn1$^{154Q/2Q}$* KI mice that there are some common, and largely unique, biological pathways overrepresented in the inferior olive and cerebellum SCA1 transcriptomics (*Figure 4B*).

We surveyed our entire ATXN1-82Q Tg cerebellar and inferior olive transcriptomics dataset for Cic binding site DNA motifs at 12 weeks of age, and found significant enrichment for Cic binding sites within the promoter region for up- and down-regulated ATXN1-82Q Tg cerebellar genes (*Table 3*). Interestingly, weak but significant enrichment for one Cic DNA motif was also identified among down-regulated ATXN1-82Q Tg inferior olive genes at 12 weeks of age (*Table 3*). Considering that polyQ-expanded ATXN1 is not expressed in inferior olive of ATXN1-82Q Tg mice (*Figure 5A and B*), this finding suggests that Cic may also function in the inferior olive transcriptome independent of mutant ATXN1 or dependent on endogenous Atxn1.

## Genes are altered in opposing directions in the cerebellum of two SCA1 mouse models

Enriched GO annotations for the ATXN1-82Q Tg cerebellum appear similar to the *Atxn1$^{154Q/2Q}$* KI clusters, and include nervous system development, cell-cell signaling, system process, and response to organic substance (*Figures 4B* and *6D*). However, closer inspection of the gene expression differences suggested that while similar clusters are enriched among all genes altered in these tissues, the

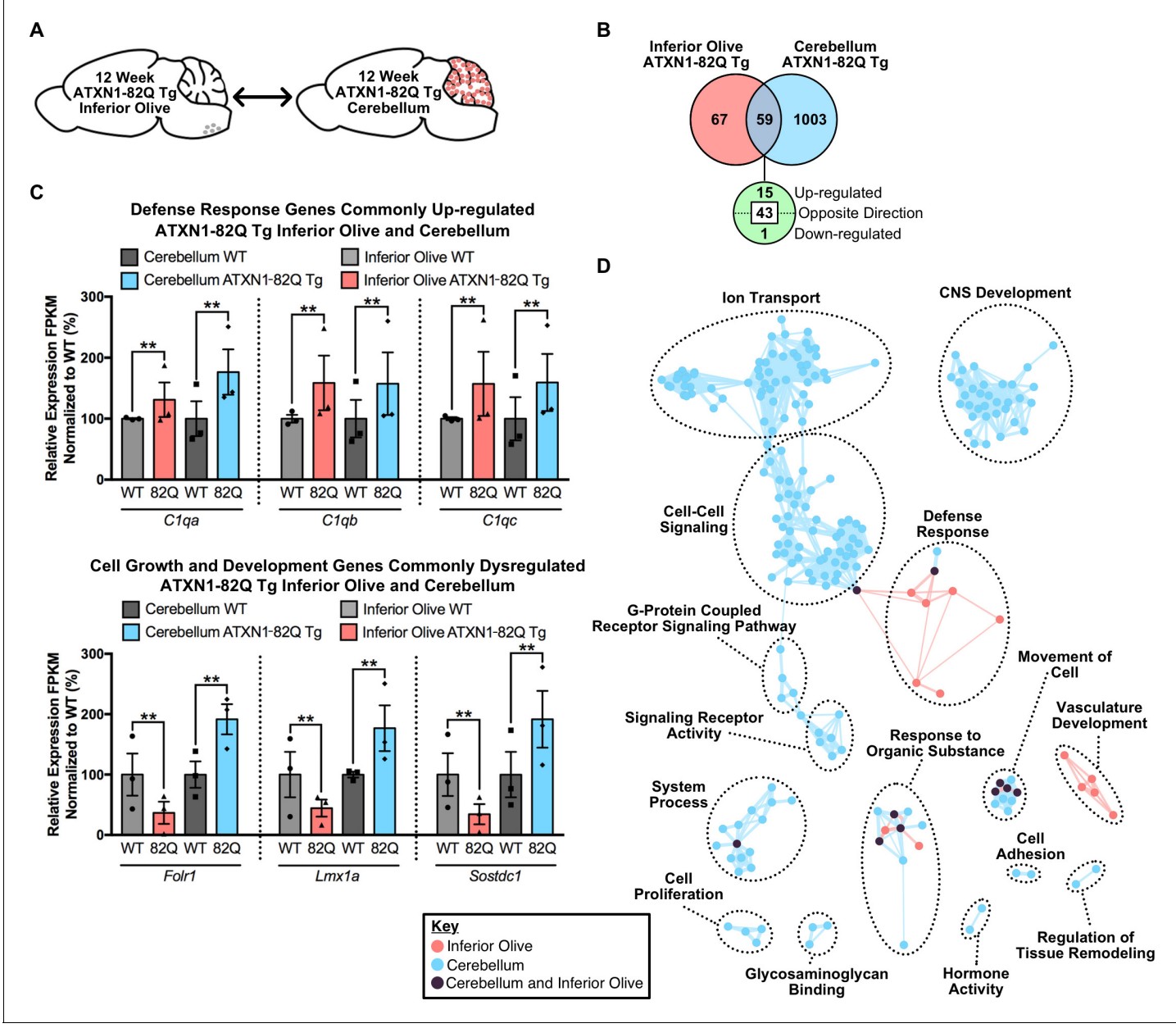

**Figure 6.** Divergent gene expression changes and low biological pathway overlap between 12 week old ATXN1-82Q Tg inferior olive and cerebellum. (A) Schematic of the cross-tissue comparison in ATXN1-82Q Tg mice. (B) Total number of differentially regulated genes that are common, and uniquely altered, in the inferior olive and cerebellum of 12 week old ATXN1-82Q Tg mice (FDR p-value < 0.05; n = 3 males per genotype). (C) A subset of genes commonly up-regulated in the inferior olive and cerebellum, and genes altered in opposing directions (** FDR p-value < 0.01; n = 3 males per genotype for RNA-seq). (D) Clustering of GO terms from the ATXN1-82Q Tg inferior olive and cerebellum differentially expressed genes lists.

DOI: https://doi.org/10.7554/eLife.39981.039

The following source data and figure supplements are available for figure 6:

**Source data 1.** 5 week old ATXN1-82Q Tg cerebellum functional annotation enrichment analysis.
DOI: https://doi.org/10.7554/eLife.39981.044
**Source data 2.** 12 week old ATXN1-82Q Tg functional annotation clustering enrichment analysis.
DOI: https://doi.org/10.7554/eLife.39981.045
**Figure supplement 1.** Chemical Synaptic Transmission and Regulation of Cell Communication are the most enriched pathways in the 5 week ATXN1-82Q Tg cerebellum.
DOI: https://doi.org/10.7554/eLife.39981.040
**Figure supplement 2.** Pathways related to Synaptic Signaling are enriched in the 12 week ATXN1-82Q Tg cerebellum.

*Figure 6 continued on next page*

*Figure 6 continued*

DOI: https://doi.org/10.7554/eLife.39981.041

**Figure supplement 3.** Common gene expression changes and pathway enrichment in the 5 and 12 week ATXN1-82Q Tg cerebellum.

DOI: https://doi.org/10.7554/eLife.39981.042

**Figure supplement 4.** Divergent gene expression changes and pathway enrichment in the 5 week ATXN1-82Q Tg inferior olive and cerebellum.

DOI: https://doi.org/10.7554/eLife.39981.043

directionality of the gene expression changes can vary between mouse models. As a result, we compared the molecular changes between these two mouse models in the cerebellum (*Figure 7A*).

Of the genes altered in each tissue, a total of 316 genes overlapped (*Figure 7B*). The directionality of these changes was largely discordant between mouse models, with 177 genes altered in opposing directions (*Figure 7B and C*). The majority of these discordant genes were down-regulated in the *Atxn1*$^{154Q/2Q}$ KI cerebellum and up-regulated in the ATXN1-82Q Tg cerebellum (*Figure 7C*). Among the commonly up-regulated genes, three of the 13 genes were associated with the regulation of cell death (*Figure 7C and D*). Among the genes commonly down-regulated, which consisted of 126 annotated genes, a subset were associated with neuron projection and development (*Figure 7C and D*). There were a total of 151 genes down-regulated in *Atxn1*$^{154Q/2Q}$ KI and up-regulated in ATXN1-82Q Tg, and many were linked to nervous system development (*Figure 7C and D*). Synaptic signaling-related genes were largely up-regulated in the *Atxn1*$^{154Q/2Q}$ KI and down-regulated in ATXN1-82Q Tg (*Figure 7D*). This suggests that either the level of mutant ATXN1 protein expression, the length of the polyQ repeat, or the cell-type expressing the pathogenic protein may impact gene expression directionality.

## Discussion

Neurodegeneration can affect multiple brain regions, however, many neurodegenerative studies have focused their research extensively on one classically defined affected tissue. Due to the

**Table 3.** Enrichment for Cic binding site DNA motifs in down- and up-regulated genes from 12 week old ATXN1-82Q Tg inferior olive and cerebellum.

| Cic DNA motif | Brain region | Motif occurrence | Down-Regulated | Shuffled | p-value | q-value |
|---|---|---|---|---|---|---|
| TGAATGAA | Inferior Olive | Present | 12 | 1 | 0.002 | 0.002 |
| | | Absent | 56 | 67 | | |
| | Cerebellum | Present | 90 | 48 | 1.9e-4 | 4.0e-4 |
| | | Absent | 510 | 552 | | |
| TGAATGGA | Inferior Olive | Present | 19 | 13 | 0.312 | 0.187 |
| | | Absent | 49 | 55 | | |
| | Cerebellum | Present | 110 | 86 | 0.073 | 0.061 |
| | | Absent | 490 | 514 | | |
| Cic DNA Motif | Brain Region | Motif Occurrence | Up-Regulated | Shuffled | p-value | q-value |
| TGAATGAA | Inferior Olive | Present | 10 | 7 | 0.606 | 0.318 |
| | | Absent | 60 | 63 | | |
| | Cerebellum | Present | 97 | 51 | 4.9e-5 | 2.1e-4 |
| | | Absent | 366 | 412 | | |
| TGAATGGA | Inferior Olive | Present | 15 | 7 | 0.103 | 0.072 |
| | | Absent | 55 | 63 | | |
| | Cerebellum | Present | 96 | 59 | 0.002 | 0.002 |
| | | Absent | 367 | 404 | | |

FDR p-value calculated using the two-stage linear step-up procedure of Benjamini, Krieger, and Yekutieli (Q 5%).

DOI: https://doi.org/10.7554/eLife.39981.046

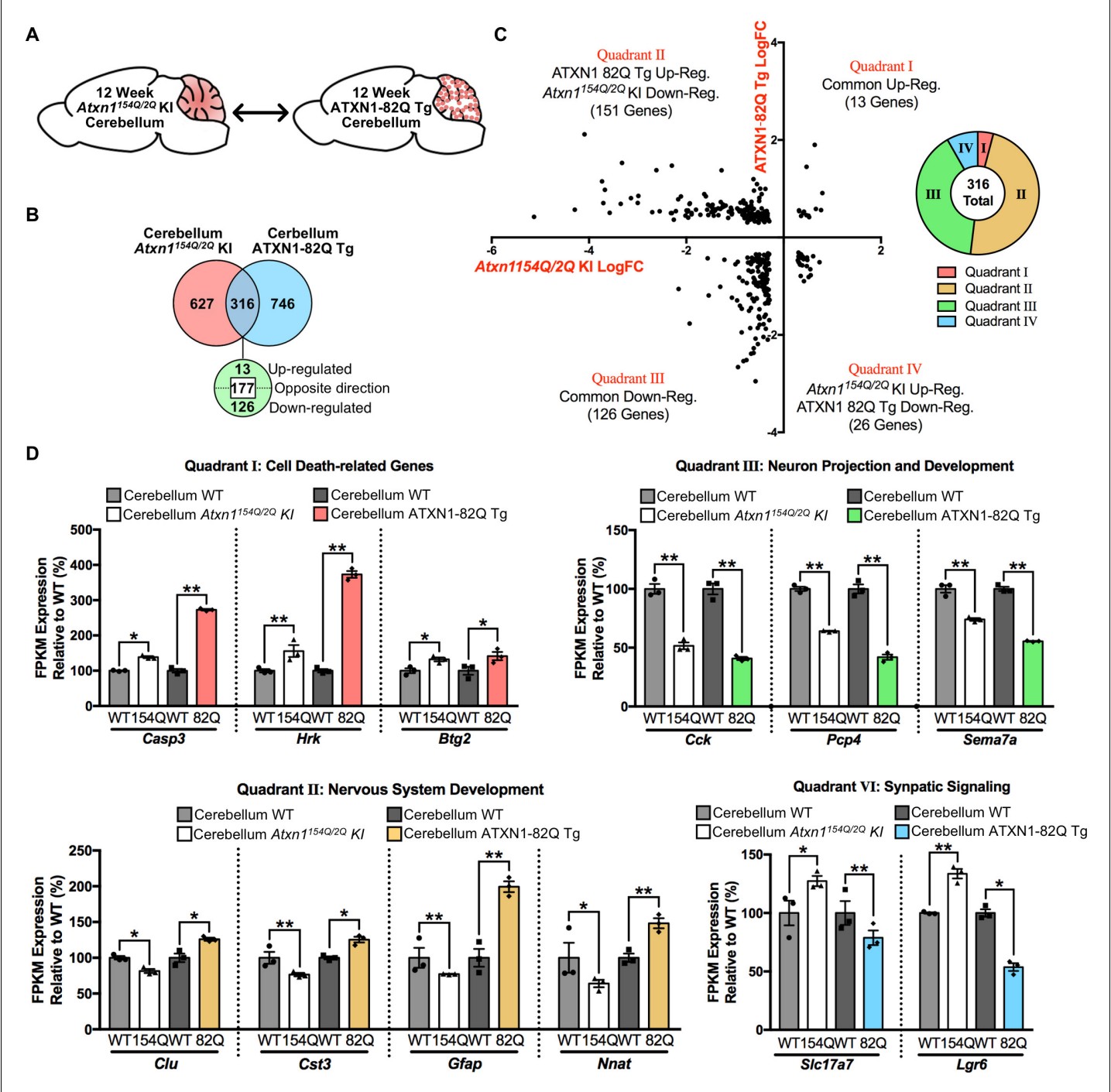

**Figure 7.** Genes commonly regulated in the cerebellum across 12 week old SCA1 mouse models are altered in opposing directions. (**A**) Schematic of the cross-model comparison in 12 week old $Atxn1^{154Q/2Q}$ KI and ATXN1-82Q Tg cerebellum. (**B**) Number and directionality of shared differentially regulated genes in the $Atxn1^{154Q/2Q}$ KI and ATXN1-82Q Tg cerebellum (FDR p-value < 0.05; n = 3 males per genotype for RNA-seq). (**C**) Log fold change for genes commonly altered in both the $Atxn1^{154Q/2Q}$ KI and ATXN1-82Q Tg cerebellum, and the proportion of commonly altered genes within each quadrant. (**D**) Common genes up-regulated or down-regulated in the cerebellum from both mouse models (Quadrant I and III), and genes regulated in opposing directions (Quadrant II and IV) at 12 weeks of age (* FDR p-value < 0.05; ** FDR p-value < 0.01; n = 3 males per genotype for RNA-seq).

DOI: https://doi.org/10.7554/eLife.39981.047

increasing evidence that there are underlying differences in tissue-specific cell populations and responses to external stimuli, it is necessary to begin examining all affected and sparred tissues in neurodegenerative conditions to determine their underlying similarities and differences, which remain unknown. Identifying commonalities between tissues will provide insight into potential therapeutic targets, while elucidating discrepancies between tissues will provide a more thorough understanding of how disease uniquely affects each tissue. Further, previous work has highlighted that different mouse models created to study specific neurodegenerative diseases have somewhat different molecular alterations, sometimes making it difficult to discern a common mechanism driving pathological and clinical phenotypes (*Hargis and Blalock, 2017*). We addressed these questions for the first time in a SCA1 context by 1) identifying underlying molecular alterations that occur in the inferior olive, 2) assessing how these molecular alterations change during early SCA1 disease progression, 3) highlighting prevalent similar and unique biological pathways in two affected SCA1 tissues, and 4) conducting a cross-model comparison between two SCA1 mouse modes (*Figure 8*).

Response to organic substance and hormone-related pathways were significantly enriched in both the *Atxn1^{154Q/2Q}* KI and ATXN1-82Q Tg inferior olive at 5 weeks of age, which coincides with the earliest reported onset of behavioral phenotypes (*Figure 2—figure supplement 1*; *Figure 5—*

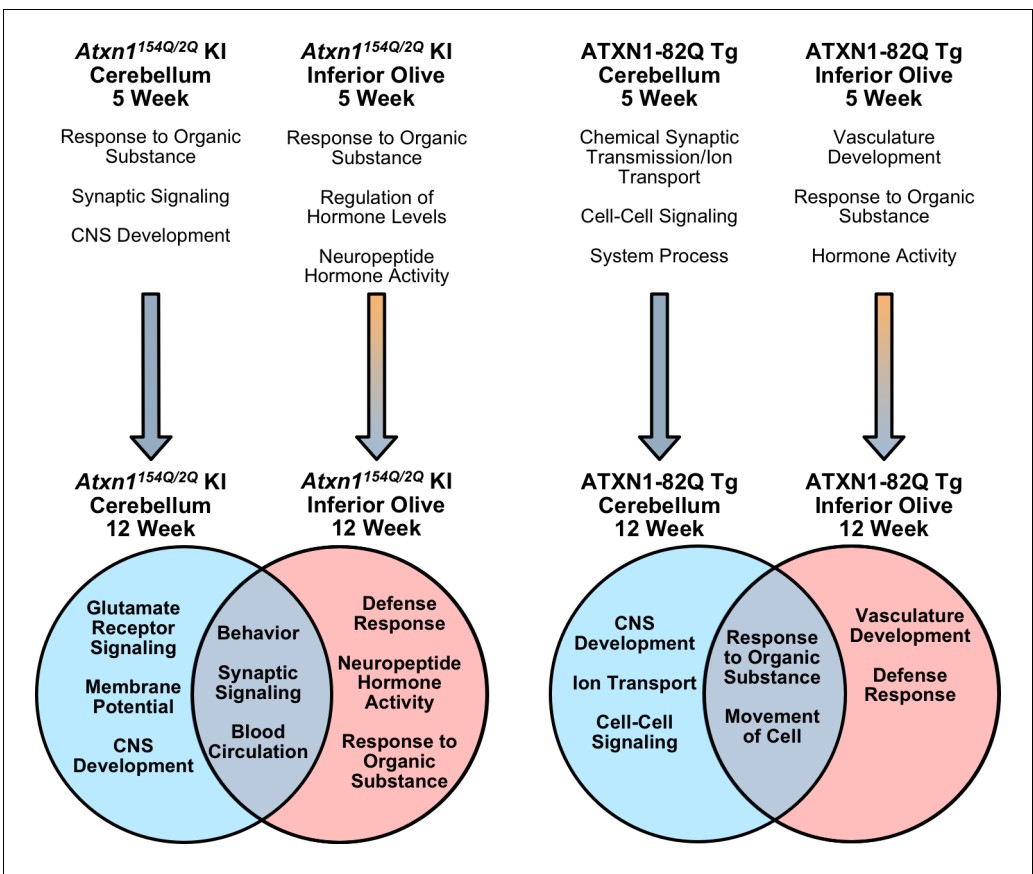

**Figure 8.** Summary of biological and molecular pathway enrichment across SCA1 mouse models in each brain region over time. Differentially regulated genes in the 5 week old inferior olive and cerebellum from both SCA1 mouse models are associated with specific biological and molecular pathways, a subset of which are listed. The 5 week old *Atxn1^{154Q/2Q}* KI cerebellum pathways are derived when using a nominal p-value < 0.01. In the cerebellum, the majority of differentially regulated genes are down-regulated at both the 5 and 12 week timepoints (blue arrow). In the SCA1 inferior olive, the majority of genes are up-regulated at 5 weeks of age, which is followed by a shift toward gene down-regulation (orange and blue gradient arrow). In SCA1 mice at 12 weeks of age, there are enriched biological and molecular pathways that are both conserved, and unique, to specific affected tissues.

DOI: https://doi.org/10.7554/eLife.39981.048

*figure supplement 1*; *Figure 8*). A subset of the *Atxn1^{154Q/2Q}* KI response to organic substance pathway contained ISGs, which are known to contribute to immune-related processes (*Figure 2—figure supplement 1D*). A small subset of immune-related genes were also identified in the 5 week ATXN1-82Q Tg inferior olive (*Figure 5—figure supplement 1D*). This suggests that some, but not a significant amount, of Defense Response-related genes are significantly different in the *Atxn1^{154Q/2Q}* KI and ATXN1-82Q Tg inferior olive at 5 weeks of age. However, by 12 weeks of age, both the *Atxn1^{154Q/2Q}* KI and ATXN1-82Q Tg inferior olive differentially regulated gene lists are made up of a substantial portion of defense response, or immune-related, genes (*Figures 2D*, *5E* and *8*; *Figure 5—figure supplement 2C*). The biological and pathological significance of the defense response in the SCA1 inferior olive will require further examination to determine whether this is a component of a pathological pathway or protective response. One interesting finding in this study is that although the defense response was a top significantly enriched pathway in the inferior olive of two distinct SCA1 models, the genes differentially regulated in these models do not strongly overlap (*Figure 5F and G*). For example, ISGs such as the retinoic acid-inducible genes (RIG-1) like receptors *Ddx58* (RIG-1) and *Ifih1* (MDA5) are significantly up-regulated in the *Atxn1^{154Q/2Q}* KI inferior olive, but do not change in the ATXN1-82Q Tg inferior olive. Due to the fundamental difference in the expression of polyQ-expanded mutant ATXN1 protein in the inferior olive between the two different SCA1 models (*Figures 2A*, *5A and B*), the molecular and cellular mechanisms leading to the increased defense response will be different.

Enrichment for defense response-related genes in the 12 week old ATXN1-82Q Tg inferior olive is particularly striking given that ATXN-82Q is not detectable at the mRNA and protein level in this tissue compared to the cerebellum (*Figure 5A and B*). Further, polyQ-expanded ATXN1 would only be expressed in neurons if it is expressed below detectable levels since the expression of ATXN1-82Q is driven by the *Pcp2* promoter (*Burright et al., 1995*). Therefore, the large number of defense response-related genes, especially those expressed at higher levels in glia cells, suggest that some non-cell autonomous effects on transcription are occurring in ATXN1-82Q Tg inferior olive. Interestingly, potential upstream regulators identified with IPA did not indicate that Irf7 was a significant regulatory factor in the ATXN1-82Q Tg mice (*Supplementary File 5*). This further enhances our prediction that polyQ-expanded mutant Atxn1 may directly influence the Irf7 activity in a cell-autonomous manner in glia cells in the *Atxn1^{154Q/2Q}* KI inferior olive (*Figure 2I*). While alterations in glia-expressed genes appear to be a major component of the inferior olive transcriptome, we cannot rule out the possibility that some of these gene expression alterations are occurring in other cell-types in the inferior olive. One alternative explanation for the gene expression changes in the ATXN1-82Q Tg inferior olive is dysfunction and pathology of cerebellar PCs having a retrograde effect on the inferior olive.

Our results suggest that gene expression changes are dynamic between 5 and 12 weeks of age in the inferior olive of both mouse models (*Figure 8*). Comparison of the differentially regulated gene lists over time found a large proportion of genes altered at both time-points with dysregulated expression (*Figure 2—figure supplement 4B and C*; *Figure 5—figure supplement 3B and C*). These genes were typically up-regulated at 5 weeks of age, but were down-regulated at 12 weeks of age, and this was consistent across both SCA1 mouse models (*Figure 2—figure supplement 4B and C*; *Figure 5—figure supplement 3B and C*; *Figure 8*). This is in comparison to the cerebellar time-point comparison, which found that the majority of overlapping genes remained consistently either down-regulated or up-regulated over time (*Figure 3—figure supplements 2B, C*, *4B and C*; *Figure 6—figure supplement 3B and C*; *Figure 8*). Whether this switch in gene expression regulation is tied to disease onset and progression, pathological cerebellar input, or other mechanisms is unknown.

One caveat of the present study is the few number of differentially regulated genes identified in the 5 week old *Atxn1^{154Q/2Q}* KI cerebellum (*Figure 3—figure supplement 1B*). Previous reports from a 4 week and 7 week time-point microarray study identified more genes significantly altered in this brain region (*Crespo-Barreto et al., 2010*; *Gatchel et al., 2008*). After extending our p-value to include nominally significant genes p < 0.01, we found significant overlap between our dataset with the previously published datasets (*Figure 3—figure supplement 3A*). This suggests that animal variability, differences in animal housing, and potentially our stringent FDR p-value < 0.05 cut-off ultimately led to a small number of genes differentially regulated at 5 weeks of age. Despite this, our current findings in the *Atxn1^{154Q/2Q}* KI and ATXN1-82Q Tg cerebellum are consistent with previously

published reports (*Gatchel et al., 2008*; *Ingram et al., 2016*). In the Ingram et al. study (*Ingram et al., 2016*) investigating cerebellar molecular alterations in ATXN1-82Q Tg mice that correlate with disease progression, the 10 hub genes that correlate with disease progression in PCs were also significantly down-regulated in our 5 and 12 week datasets (*Figure 3D*; GEO accession 122099). In addition, *Cck* and *Col18a1* were down-regulated in our 5 and 12 week datasets (*Figures 4E* and *7D*; *Figure 2—figure supplement 4E*), which match the previous report (*Ingram et al., 2016*).

There was a marked difference in the biological functions associated with the differentially expressed genes in both the 12 week old $Atxn1^{154Q/2Q}$ KI and ATXN1-82Q Tg inferior olive and cerebellum comparisons (*Figures 4B*, *6D* and *8*). The defense response was consistently enriched in the inferior olive relative to the cerebellum across SCA1 mouse models (*Figures 4B*, *6D* and *8*). The lack of significantly enriched defense response-related pathways in the $Atxn1^{154Q/2Q}$ KI and ATXN1-82Q Tg cerebellum is particularly striking given that a previous study found increased gliosis in the cerebellum around this time-point (*Cvetanovic et al., 2015*). Further, our transcriptomics data found a significant decrease in *Gfap* expression in the $Atxn1^{154Q/2Q}$ KI cerebellum (*Figure 7D*), which is consistent with a recently published report (*Edamakanti et al., 2018*). This may indicate that protein levels and mRNA expression do not coincide in some instances. It is also possible that defense response-related genes are enriched among activated glia in the SCA1 cerebellum, however, those differentially regulated genes may be lost due to sequencing in bulk cerebellar lysates. There is significant up-regulation for some components of the defense response, including ISGs in the $Atxn1^{154Q/2Q}$ KI cerebellum (*Figure 4D*), and compliment cascade factors in the ATXN1-82Q Tg cerebellum (*Figure 6C*), however, the relatively small number of genes associated with the defense response that are differentially regulated within the cerebella suggests that it is not a major feature of the SCA1 cerebellar transcriptome.

Enrichment for Cic binding sites within the $Atxn1^{154Q/2Q}$ KI and ATXN1-82Q Tg cerebellum down-regulated genes (*Tables 2* and *3*) is consistent with a previous report (*Fryer et al., 2011*), indicating that the presence of polyQ-expanded ATXN1 acts synergistically with Cic to further repress transcriptional activity. While there was a degree of enrichment for Cic binding sites among the up-regulated $Atxn1^{154Q/2Q}$ KI and down-regulated ATXN1-82Q Tg inferior olive genes, it was not the most significantly over-represented potential transcriptional regulator at the 12 week time-point (*Tables 1–3*; *Figure 2—source data 3*; *Supplementary File 5*). While we cannot rule out the role of Cic-Atxn1 interactions on regulating molecular alterations in the inferior olive, other transcriptional regulators, such as Irf7 and Esr1, may be regulating a proportion of the differentially regulated genes in the inferior olive in the $Atxn1^{154Q/2Q}$ KI and ATXN1-82Q Tg mouse models.

The two SCA1 mouse cerebellar datasets had significant enrichment for similar biological functions, which initially suggested to us that similar genes may be altered in these two mouse models despite the mouse model differences in polyQ-expansion, cell-type expression, expression level of polyQ-expanded ATXN1, human vs. mouse protein, or genetic backgrounds (*Burright et al., 1995*; *Watase et al., 2002*). While a proportion of genes did overlap between these two mouse models, we observed a difference in the directionality of gene expression changes across the models (*Figure 7B and C*). A similar observation can be found in SCA2 transgenic and KI mouse models (*Halbach et al., 2017*; *Pflieger et al., 2017*). The majority of the commonly dysregulated genes were up-regulated in the ATXN1-82Q Tg mouse cerebellum and down-regulated in the $Atxn1^{154Q/2Q}$ KI cerebellum (*Figure 7C*), and contained genes associated with nervous system development (*Figure 7D*). The disparity in significant gene expression directionality in these two models may be due to the cell-type-specific expression of polyQ-expanded ATXN1. The PC-specific expression of ATXN1-82Q, and its effect on PC function, may lead to up-regulation of a subset of non-neuronal genes. However, cell-type appropriate expression of polyQ-expanded Atxn1 in $Atxn1^{154Q/2Q}$ KI cerebellum may lead to alterations in similar genes via cell-type specific intrinsic mechanisms. Despite the discrepancy in the directionality of the differentially expressed genes, previous studies show that the two mouse models exhibit similar behavioral and pathological phenotypes. Both models manifest motor coordination deficits at approximately 5 weeks of age (*Clark et al., 1997*; *Watase et al., 2002*). Molecular layer thinning is obvious at 12 weeks in the ATXN1-82Q Tg mouse model, but occurs at a later time-point in the $Atxn1^{154Q/2Q}$ KI mouse (*Barnes et al., 2011*; *Clark et al., 1997*; *Duvick et al., 2010*; *Watase et al., 2002*).

Variability in gene expression in the brain across multiple inbred mouse strains has previously been demonstrated (*Nadler et al., 2006*). The two SCA1 mouse models analyzed in the present study were maintained on different mouse genetic backgrounds. The $Atxn1^{154Q/2Q}$ KI mice are maintained on a C57BL/6J background, while the ATXN1-82Q Tg are kept on a pure FVB/NJ background. Therefore, the genetic interaction between polyQ-expanded ATXN1 and the mouse genetic background should be considered. In addition, the two SCA1 mouse models utilize either the mouse or human homologue of polyQ-expanded ATXN1. The amino acid sequence similarity between the mouse and human ATXN1 is 89% when excluding the polyQ-tract. While this indicates a high degree of sequence similarity, how the sequence differences influence subsequent gene expression changes is unknown.

In summary, our work shows for the first time that there are specific biological pathways enriched in a less studied SCA1 affected tissue, the inferior olive. Further, there are intrinsic molecular differences between two affected tissues in SCA1, and biological pathways that are largely brain-region-specific are conserved across SCA1 mouse models, regardless of the expression of polyQ-expanded ATXN1. The commonalities between affected tissues indicate that a subset of molecular changes may be useful as disease biomarkers in SCA1.

# Materials and methods

## Key resources table

| Reagent type | Designation | Source or reference | Identifiers | Additional information |
|---|---|---|---|---|
| Strain, or strain background (*Mus musculus*, C57BL/6J) | $Atxn1^{154Q/2Q}$ KI | PMID:12086639 | RRID:MGI:2429435, backcrossed in house on C57BL/6J background | |
| Strain, or strain background (*Mus musculus*, FVB/NJ) | ATXN1-82Q Tg | PMID:7553854 | RRID:MGI:2447854 backcrossed in house on FVB/NJ background | |
| Strain, or strain background (*Mus musculus*, C57BL/6J) | C57BL/6J | The Jackson Lab | Stock: 000664 | |
| Strain, or strain background (*Mus musculus*, FVB/NJ) | FVB/NJ | The Jackson Lab | Stock: 001800 | |
| Cell line (*Mus musculus*) | BV2 | A kind gift from Dr. Katerina Akassoglou (*Adams et al., 2007*) | RRID: CVCL_0182 | |
| Transfected construct | human Flag-ATXN1 82Q | A kind gift from Dr. Huda Zoghbi | Addgene Plasmid #33237 | |
| Transfected construct | mammalian expression vector pcDNA3.1+ | Invitrogen (ThermoFisher Scientific) | Cat. # V790-20 | |
| antibody | rabbit polyclonal anti-Atxn111750 | a kind gift from Huda Zoghbi (*Servadio et al., 1995*) | RRID:AB_2721278 | (1:1,000) WB |
| Antibody | mouse monoclonal anti-Calbindin | Sigma | Cat. #: C9848 RRID:AB_476894 | (1:2,000) IHC |
| Antibody | mouse monoclonal anti-Gapdh | Sigma | Cat. #G8795 RRID:AB_1078991 | (1:10,000) WB |
| Antibody | chicken polyclonal anti-Gfap | Abcam | Cat. # ab4674 RRID:AB_304558 | (1:1,000) IHC |
| Antibody | rabbit polyclonal anti-Iba1 | Wako | Cat. # 019–19741 RRID:AB_839504 | (1:2,000) IHC |

*Continued on next page*

*Continued*

| Reagent type | Designation | Source or reference | Identifiers | Additional information |
|---|---|---|---|---|
| Antibody | goat polyclonal anti-chicken Alexa555 | ThermoFisher | Cat. #A21437 RRID:AB_2535858 | (1:500) IHC |
| Antibody | goat polyclonal anti-mouse Alexa488 | ThermoFisher | Cat. # A11001 RRID:AB_2534069 | (1:500) IHC |
| Antibody | goat polyclonal anti-rabbit Alexa568 | ThermoFisher | Cat. #A11011 RRID:AB_143157 | (1:500) IHC |
| Antibody | donkey anti-rabbit | GE Healthcare | Cat. #NA934 RRID:AB_772206 | (1:4,000) WB |
| Antibody | anti-mouse | GE Healthcare | Cat. #NXA931 RRID:AB_772209 | (1:4,000) WB |
| Commercial assay or kit | RNeasy Mini Kit | Qiagen | Cat #74106 | |
| Commercial assay or kit | iScript cDNA Synthesis Kit | Bio-Rad | Cat. #1708891 | |
| Commercial assay or kit | iTaq Universal Probes Supermix | Bio-Rad | Cat. #1725131 | |
| Commercial assay or kit | Pierce BCA Protein Assay | ThermoFisher | Cat. #23225 | |
| Commercial assay or kit | Amaxa Cell Line Nucleo fector Kit T | Lonza | Cat. #VCA-1002 | |
| Other | DAPI | Vector Laboratories | Cat. #H-1500 RRID:AB_2336788 | |
| Other | *Actb* probe | Applied Biosystems | Cat. #4352932E | |
| Other | *ATXN1* probe | Applied Biosystems | Cat. # 4331182 Hs00165656_m1 | |
| Other | *Calb1* probe | Applied Biosystems | Cat. #4331182 Mm00486647_m1 | |
| Other | *Gapdh* probe | Applied Biosystems | Cat. #4352933E | |
| Other | *Ifitm3* probe | Applied Biosystems | Cat. # 4331182 Mm00847057_s1 | |
| Other | *Ifr7* probe | Applied Biosystems | Cat. #4331182 Mm00516793_g1 | |
| Other | *Oasl2* probe | Applied Biosystems | Cat. # 4331182 Mm01201449_m1 | |
| Other | *Stat1* probe | Applied Biosystems | Cat. # 4331182 Mm01257286_m1 | |
| Software, algorithm | Ingenuity Pathway Analysis (IPA) (Spring Release 2018) | http://www.ingenuity.com/products/pathways_analysis.html | RRID:SCR_008653 | |
| Software, algorithm | NIH DAVID v6.8 | http://david.abcc.ncifcrf.gov/ | RRID:SCR_001881 | |
| Software, algorithm | TopHat v2.1.0 | http://tophat.cbcb.umd.edu/ | RRID:SCR_013035 | |
| Software, algorithm | Cufflinks v2.2.1 | http://cole-trapnell-lab.github.io/cufflinks/ | RRID:SCR_014597 | |
| Software, algorithm | Bowtie2 v2.1.0 | http://bowtie-bio.sourceforge.net/bowtie2/index.shtml | RRID:SCR_016368 | |
| Software, algorithm | FastQC v0.11.3 | http://www.bioinformatics.babraham.ac.uk/projects/fastqc/ | RRID:SCR_014583 | |
| Software, algorithm | Cummerbund | http://compbio.mit.edu/cummeRbund/index.html | RRID:SCR_014568 | |

*Continued on next page*

*Continued*

| Reagent type | Designation | Source or reference | Identifiers | Additional information |
|---|---|---|---|---|
| Software, algorithm | Bioconductor R with heatmap.2 gplots package | https://www.rdocumentation.org/packages/gplots/versions/3.0.1/topics/heatmap.2 | | |
| Software, algorithm | R project for statistical computing v3.3.3 | http://www.r-project.org/ | RRID:SCR_001905 | |
| Software, algorithm | MEME Suite - Motif-based sequence analysis tools | http://meme-suite.org/ | RRID:SCR_001783 | |
| Software, algorithm | EnrichmentMap | http://baderlab.org/Software/EnrichmentMap | RRID:SCR_016052 | |
| Software, algorithm | Cytoscape v3.5.1 | http://cytoscape.org | RRID:SCR_003032 | |
| Software, algorithm | GraphPad Prism | http://www.graphpad.com/ | RRID:SCR_002798 | |
| Software, algorithm | ImageJ | https://imagej.net/ | RRID:SCR_003070 | |

## Mice

All animal care procedures were approved by the Yale University Institutional Animal Care and Use Committee. Mice were kept in a 12 hr light/dark cycle with standard chow and *ad libitum* access to water. *Atxn1*$^{154Q/2Q}$ KI mice were maintained on a pure C57BL/6J background, and the ATXN1-82Q Tg mice on a FVB/NJ background (*Burright et al., 1995*). For RNA-sequencing, males at 5 and 12 weeks of age were used. For immunohistochemistry and western blotting, a combination of males and females were used.

## RNA extraction and RNA-sequencing

Cerebella and inferior olive were dissected out, flash frozen in liquid nitrogen, and stored at −80°C until processing. For inferior olive, the brainstem was removed and the inferior olive dissected using the decussation of the pyramid and pons as a reference (*Yu et al., 2014*). The location of the dissection was verified under a dissection microscope before flash freezing. To eliminate variability in dissection, the same individual performed the macro-dissections of the cerebellum and inferior olive.

RNA was extracted using the Qiagen RNeasy Mini Kit, and DNA was removed with DNase I as described in the manufacturer's protocol. Total RNA was sent to the Yale Center for Genome Analysis for processing. RNA integrity was measured with capillary electrophoresis (Agilent BioAnalyzer 2100, Agilent Technologies) and RIN values were checked to ensure RNA integrity before proceeding with RNA-seq (*Supplementary File 6*). Libraries were generated using oligo-dT purification of poly-adenylated RNA, which was then reverse transcribed into cDNA that was fragmented, blunt ended, and ligated to adaptors. Base pair size was determined with capillary electrophoresis, and the library was quantified before pooling and sequencing on a HiSeq 2000 using a 75 bp paired-end read strategy. Sequencing data can be accessed under GEO accession # (GSE122099).

## RNA-sequencing data analysis

TopHat2 v2.1.0 was utilized to align reads to the mouse reference genome (mm10) before quantification and differential expression analysis with cufflinks v2.2.1 (*Kim et al., 2013a*; *Roberts et al., 2011*; *Trapnell et al., 2012*; *Trapnell et al., 2010*). Cuffnorm was utilized for generating normalized expression values (*Trapnell et al., 2010*). Annotations with a FDR adjusted p-value (q < 0.05) were considered significant, and genes with less than 0.5 FPKM values in one biological group were not considered in the final analysis. For the 5 week *Atxn1*$^{154Q/2Q}$ KI cerebellum, a nominal p-value < 0.01 was used for analysis (*Figure 3—figure supplements 3* and *4*; *Figure 4—figure supplement 1*). Sample information and quality control was visualized using CummeRbund in R v3.3.3 (*Goff et al., 2013*). Enrichment analysis was conducted using NIH DAVID 6.8 using the GO_TERM FAT and

KEGG annotations, with measurable genes detected in our RNA-sequencing experiments serving as the reference background (*Huang et al., 2009a*; *Huang et al., 2009b*). NIH DAVID functional annotation clustering parameters included a minimum of five initial group and final group membership, which preferentially produces larger group sizes. Cluster names on the bar graphs were named based on the most significantly enriched biological pathway in the cluster, unless otherwise specified. The rationale for using the FAT annotations was that it contains a filtered version of the GOTERM ALL category, which includes all GO annotations from level 1 (least specific) to level 5 (most specific). All enriched GO annotations with a FDR p-value cut-off of 0.05 and a minimum of 5 genes in each cluster were included in the analysis. For the 5 week old $Atxn1^{154Q/2Q}$ KI cerebellum, a nominal p-value < 0.01 was used for enrichment analysis cut-off (*Figure 3—figure supplement 4D*; *Figure 4—figure supplement 1C*). Networks were visualized using EnrichmentMap in Cytoscape v3.5.1 (*Merico et al., 2010*). GO annotations that did not relate to neuronal function (for example, terms related to muscle function), were excluded from the EnrichmentMap. Edge width corresponds with the correlation coefficient between nodes, and the similarity cutoff overlap coefficient was set to 0.8. This strict coefficient cutoff allowed for the formation of clusters of nodes with similar GO annotations. Qiagen's Ingenuity Pathway Analysis (Spring Release 2018) was utilized to identify enriched canonical pathways predicated to be activated or inhibited. Benjamini-Hochberg adjusted p-values were used for identifying enriched pathways. For analyzing upstream regulators, the following parameters were utilized: an activation z-score of absolute value (2+), p-value overlap of 0.01, depth of 2 and breadth of 5, and only direct relationships were included. Heatmaps were generated using normalized read counts and the package 'gplots' in R v3.3.3. Prism v7 was used for generating barplots of RNA-seq read counts.

## Immunofluorescence staining

Mice were anesthetized before intracardial perfusion with phosphate buffer saline (PBS) and 4% paraformaldehyde (PFA). Brains were extracted and fixed overnight in 4% PFA before incubation in 20% and 30% sucrose gradient. Samples were frozen in OCT compound (VWR, 4583) and sliced in 30 µm sections on a cryostat. Sections were washed in PBS and PBS with 0.1% Triton-X before incubation in 5% normal goat serum (Jackson Labs 005-000-121) at room temperature. Primary antibody incubation was carried out at 4°C with the following antibodies: mouse anti-Calbindin, rabbit anti-Iba1, and chicken anti-Gfap. Sections were washed before incubation in secondary antibodies goat anti-mouse Alexa488, goat anti-rabbit Alexa568, and goat anti-chicken Alexa555. Sections were mounted and coverslipped with DAPI. Fluorescent images were scanned on a Zeiss LSM800 confocal microscope. Three brain slices were imaged and quantified for each mouse.

## Cell counting and fluorescence intensity

ImageJ (National Institutes of Health) was used for all image processing. To measure the fluorescence intensity for Gfap-positive staining, the inferior olive was traced based on calbindin-positive staining, and the area in µm was measured and mean fluorescence intensity measured. To count the total number of Iba1-positive cells, 'analyze particle' was used and manually checked by a blinded individual to ensure Iba1-positive staining co-localized with DAPI.

## Western blotting

Inferior olive and cerebella were homogenized in buffer (0.5% NP-40, 0.5% Triton-X, 0.1% SDS, 20 mM Tris pH 8.0, 180 mM NaCl, 1 mM EDTA and Roche complete protease inhibitor cocktail). Samples were then sonicated to ensure breakdown of protein aggregates before rotation at 4°C for 10 min and centrifugation for 10 min at 13,000 rpm at 4°C. Supernatant was quantified using a BCA assay (ThermoFisher 23225) and protein from each sample was added to a 8% Tris-HCl gel for western blotting. For $Atxn1^{154Q/2Q}$ KI inferior olive and cerebellum, 70 µg and 20 µg of protein were added per lane, respectively. For ATXN1-82Q Tg inferior olive and cerebellum, 10 µg of protein was added per lane. Protein was transferred onto 0.42 µm nitrocellulose, and blotted with primary antibodies rabbit anti-Atxn1 11750 (a kind gift from Dr. Huda Zoghbi) (*Servadio et al., 1995*), and mouse anti-Gapdh. Nitrocellulose was washed before incubation in secondary anti-rabbit and anti-mouse, incubated in chemiluminescence substrate (Perkin Elmer NEL103E001) and developed on a Konica SRX-101A tabletop X-ray film processor. Images were quantified using ImageJ.

## Cell culture

Murine microglial BV2 cells were a kind gift of Dr. Katerina Akassoglou (*Adams et al., 2007*). Microglial BV2 cells were cultured in DMEM with 10% heat-inactivated FBS, and maintained at 37°C with 5% $CO_2$. Cells were transfected with *pcDNA3.1* and *Flag-ATXN1-85Q* constructs using an Amaxa Nucleofector (Lonza) and the Amaxa Cell Line Nucleofector Kit T (Lonza) according to manufacturer's instructions. Cells were maintained post-nucleofection for 48 hr before RNA extraction using the Qiagen RNeasy Mini Kit and subsequent real-time quantitative reverse transcription polymerase chain reaction outlined below.

The microglia-like nature of BV2 cells was verified via immunostaining with Iba1 and the induction of cytokine production following lipopolysaccharide stimulation. BV2 cells were tested for mycoplasma contamination by PCR. Cells were pelleted, resuspended in nuclease free water, and lysed. PCR was conducted with 2 μL cell lysate using the following primers: Forward - 5'-GGCGAATGGG TGAGTAAC; Reverse - 5'- CGGATAACGCTTGCGACCT. PCR reaction products from BV2 cells were run on an agarose gel, and BV2 cells were found to be negative for mycoplasma.

Real-time quantitative reverse transcription polymerase chain reaction (RT-qPCR) cDNA was synthesized from 100 ng of total RNA from cell lines or mouse models using the iScript cDNA Synthesis Kit following the manufacturer's protocol (Bio-Rad). Quantitative real-time PCR was run on a C1000 Thermal Cycler (Bio-Rad) using probes for *ATXN1, Irf7, Ifitm3, Oasl2, Calb1, Actb,* and *Gapdh* (Applied Biosystems). All samples were run in triplicate and normalized to the *Gapdh* and *Actb* expression values for ΔCT calculation and subsequent data analysis.

## Data analysis

For Gfap- and Iba1-positive fluorescence intensity and cell counting, fluorescence intensity or total cells counted were normalized to the total area measured to attain an intensity value or cell count value per 1 $mm^2$, then normalized to control samples to obtain a percentage. Immunohistochemistry and western blotting data were analyzed using *t-test*s in Prism v7. For RT-qPCR, the Bio-Rad CFX Manager v3.1 was utilized for statistical analysis, and data were plotted in Prism v7.

## Acknowledgements

We thank Erica Ryke, Chris Castaldi, and Shrikant Mane from the Yale Center for Genome Analysis for conducting RNA-seq, Dr. Katerina Akassoglou for BV2 cells, and members from the Lim laboratory for thoughtful comments and critiques. This work was funded by the National Institute of Neurological Disorders and Stroke grants R01 NS083706 and R01 NS088321 to Janghoo Lim.

## Additional information

### Funding

| Funder | Grant reference number | Author |
| --- | --- | --- |
| National Institute of Neurological Disorders and Stroke | NS083706 | Janghoo Lim |
| National Institute of Neurological Disorders and Stroke | NS088321 | Janghoo Lim |

The funders had no role in study design, data collection and interpretation, or the decision to submit the work for publication.

### Author contributions

Terri M Driessen, Conceptualization, Data curation, Formal analysis, Validation, Investigation, Methodology, Writing—original draft, Writing—review and editing; Paul J Lee, Formal analysis, Validation, Writing—review and editing; Janghoo Lim, Conceptualization, Resources, Data curation, Formal analysis, Supervision, Funding acquisition, Investigation, Methodology, Project administration, Writing—review and editing

Author ORCIDs
Terri M Driessen (ID) http://orcid.org/0000-0002-9116-5497
Paul J Lee (ID) http://orcid.org/0000-0002-4009-8220
Janghoo Lim (ID) http://orcid.org/0000-0002-5331-210X

## Ethics

Animal experimentation: Animal experimentation: This study was performed in strict accordance with the recommendations in the Guide for the Care and Use of Laboratory Animals of the National Institutes of Health. All of the animals were handled according to approved institutional animal care and use committee (IACUC) protocols (#2016-11342) of the Yale University. The Yale University Institutional Animal Care and Use Committee approved all research and animal care procedures. We made every effort to minimize animal suffering.

## Decision letter and Author response

Decision letter https://doi.org/10.7554/eLife.39981.063
Author response https://doi.org/10.7554/eLife.39981.064

# Additional files

## Supplementary files

• Supplementary file 1. 5 week old $Atxn1^{154Q/2Q}$ KI inferior olive IPA canonical pathway enrichment.
DOI: https://doi.org/10.7554/eLife.39981.049

• Supplementary file 2. 5 week old $Atxn1^{154Q/2Q}$ KI inferior olive IPA predicted upstream regulators.
DOI: https://doi.org/10.7554/eLife.39981.050

• Supplementary file 3. Upstream regulators for differentially regulated 12 week old $Atxn1^{154Q/2Q}$ KI cerebellum genes.
DOI: https://doi.org/10.7554/eLife.39981.051

• Supplementary file 4. 5 week old ATXN1-82Q Tg inferior olive IPA canonical pathways.
DOI: https://doi.org/10.7554/eLife.39981.052

• Supplementary file 5. 12 week old ATXN1-82Q Tg inferior olive upstream regulators predicted in IPA.
DOI: https://doi.org/10.7554/eLife.39981.053

• Supplementary file 6. RNA integrity numbers (RIN) for samples used in RNA-seq analysis.
DOI: https://doi.org/10.7554/eLife.39981.054

• Transparent reporting form
DOI: https://doi.org/10.7554/eLife.39981.055

## Data availability

RNA sequencing data have been deposited in GEO under accession (number: 122099).

The following dataset was generated:

| Author(s) | Year | Dataset title | Dataset URL | Database and Identifier |
|---|---|---|---|---|
| Driessen TM, Lee PJ, Lim J | 2018 | Molecular pathway analysis towards understanding tissue vulnerability in spinocerebellar ataxia type 1 | https://www.ncbi.nlm.nih.gov/geo/query/acc.cgi?acc=GSE122099 | NCBI Gene Expression Omnibus, GSE122099 |

The following previously published datasets were used:

| Author(s) | Year | Dataset title | Dataset URL | Database and Identifier |
|---|---|---|---|---|
| Gatchel JR, Watase K, Thaller C, Carson JP, Jafar-Nejad P, Shaw CA, Zu T, Orr HT, Zoghbi HY | 2007 | Expression data from early symptomatic Sca1154Q/2Q and Sca7266Q/5Q knock-in cerebellum | https://www.ncbi.nlm.nih.gov/geo/query/acc.cgi?acc=GSE9914 | NCBI Gene Expression Omnibus, GSE9914 |

Ingram M, Leathley E, Henzler C, Duvick L, Yang R, Bergmann P, Zoghbi HY, Orr HT | 2016 | Cerebellar RNA-Seq from ATXN1 Transgenic Mice Reveals SCA1 Disease Progression and Protection Pathways | https://www.ncbi.nlm.nih.gov/geo/query/acc.cgi?acc=GSE75778 | NCBI Gene Expression Omnibus, GSE75778

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
