## [Decision Letter]

Thank you for submitting your article "Molecular pathway analysis towards understanding tissue vulnerability in spinocerebellar ataxia type 1" for consideration by *eLife*. Your article has been reviewed by three peer reviewers, and the evaluation has been overseen by a Reviewing Editor and a Senior Editor. The following individual involved in review of your submission has agreed to reveal his identity: Stefan Pulst (Reviewer #2).

The reviewers have discussed the reviews with one another and the Reviewing Editor has drafted this decision to help you prepare a revised submission.

Summary:

This paper provides comparative transcriptomic analysis of distinct brain regions in two different SCA1 mouse models. Importantly, this is the first characterization of the transcriptional signature of the inferior olive, which is a vulnerable brain region in SCA. These analyses reveal substantial overlap but also some differences in transcriptomic changes between knock-in and transgenic models, as may be expected. More importantly, these analyses found brain region-specific differences that are shared between the two distinct models of SCA1, suggesting different mechanisms of degeneration at work in the inferior olive and cerebellum. Interestingly, changes in genes representing the "defense response" are prominent in the inferior olive, appearing to reflect non-neural changes due to activation of glia. This work enriches our understanding of two commonly used models of SCA1 and provides valuable novel insights into the mechanisms that may contribute to regional neurodegeneration in SCA more generally.

Essential revisions:

1) It is not clear why a 12-week time point was chosen for analysis – a time point when both lines examined here are symptomatic. Ideally, a time point should be chosen that precedes behavioral and major morphological changes since cell loss may distort the transcriptional signatures. We ask the authors to consider adding analysis of an earlier, presymptomatic time point to assess which transcriptional changes are occurring first and the trajectory of the changes. Perhaps the 12-week time point was selected to correspond to prior published results (e.g. Ingram et al., 2016). It would be helpful for the authors to discuss the current findings in the context of previously published results.

2) The statement that "common behavioral and pathological phenotypes of the two mouse models may be due to dysregulation of a common subset of genes, and not entirely based on the gene expression directionality" is highly speculative and not supported by much data. This conclusion should be tempered or supported by more experimental evidence.

---

## [Author Response]

Essential revisions:1) It is not clear why a 12-week time point was chosen for analysis – a time point when both lines examined here are symptomatic. Ideally, a time point should be chosen that precedes behavioral and major morphological changes since cell loss may distort the transcriptional signatures. We ask the authors to consider adding analysis of an earlier, presymptomatic time point to assess which transcriptional changes are occurring first and the trajectory of the changes. Perhaps the 12-week time point was selected to correspond to prior published results (e.g. Ingram et al., 2016). It would be helpful for the authors to discuss the current findings in the context of previously published results.

The initial 12 week time-point was chosen for three reasons. First, the onset of transcriptional changes in the inferior olive are unknown, and an intermediate time-point was initially chosen to determine if gene expression changes are occurring in the inferior olive. Second, substantial cell-loss has not been reported in the cerebellum in both the *Atxn1^154Q/2Q^* KI and ATXN1-82Q Tg mouse models, and no obvious cellular changes were reported in the *Atxn1^154Q/2Q^* KI inferior olive outside of nuclear inclusion formation at an 18 week time-point (Watase et al., 2002). These observations suggested to us that RNA-seq analysis at 12 weeks was plausible without the consideration of distorted transcriptional alterations due to substantial cell loss. Third, previously published RNA-seq and microarrays in both mouse models have used the 12 week intermediate time-point, which would allow for potential comparison between our results and those previously published.

In consideration of the reviewer's suggestion of adding an earlier, pre-symptomatic time point to our analysis, we have added RNA-sequencing results from the *Atxn1^154Q/2Q^*KI and ATXN1-82Q Tg inferior olive and cerebellum at 5 weeks of age. This time-point coincides with the earliest reported onset of or prior to motor phenotypes in these mouse models (Clark et al., 1997; Watase et al., 2002). Bioinformatic analysis of the individual brain regions and mouse models, as well as the cross-tissue comparison were conducted on the 5 week data-set. Further, the commonalities and differences between the 5 week and 12 week datasets within each brain region were assessed. We believe that the addition of the 5 week time-point contributes substantially to our initial manuscript, and solidifies the findings that there are brain region-specific molecular alterations in SCA1.

Data pertaining to the 5 week *Atxn1^154Q/2Q^* KI and ATXN1-82Q Tg inferior olive and cerebellum analysis can be found in the following figures: Figure 2—figure supplements 1 and 4; Figure 3—figure supplements 1-4; Figure 4—figure supplement 1; Figure 5—figure supplements 1 and 3; Figure 6—figure supplements 1, 3, and 4.

The addition of the 5 week time-points led to a substantial revision of the text, however, the main findings from the 5 week time-points can be found in the Results section.

The incorporation of the 5 week time-point generated 13 additional Supplementary Data tables (Supplementary Data 1-3, 6, 8, 11-12, 14-15, 18-19, 21, and 24). To reduce the number of Supplementary Data tables, we have removed Supplementary Data tables found in the original manuscript that contained differentially regulated gene lists from each brain region. The justification for this is that the differentially regulated gene lists can be easily accessed in our processed data text files found on Gene Expression Omnibus (GEO accession number: 122099).

In consideration of the reviewer's comment about discussing our current findings in the context of previously published works, we have added this to the Discussion section. Briefly, our ATXN1-82Q Tg cerebellum datasets have consistent findings with those from a previous publication (Ingram et al., 2016). From that study, the 10 hub genes identified in the magenta module, which was significantly associated with ataxia, were also significantly down-regulated in our 5 and 12 week datasets. In addition, *Cck* and *Col18a1* were down-regulated in our 5 and 12 week datasets, which match the previous report (Ingram et al., 2016). In addition, our 5 and 12 week old *Atxn1^154Q/2Q^* KI cerebellum dataset overlaps with microarray findings from a previous study analyzing 4 week old and 9-12 week old *Atxn1^154Q/2Q^* KI mice (Gatchel et al., 2008; Crespo-Baretto et al., 2010).

2) The statement that "common behavioral and pathological phenotypes of the two mouse models may be due to dysregulation of a common subset of genes, and not entirely based on the gene expression directionality" is highly speculative and not supported by much data. This conclusion should be tempered or supported by more experimental evidence.

We agree with the reviewers that this statement is speculative and extrapolated from our data without much supporting evidence. In order to address this comment, we have removed this concept from the Discussion section. The Results section was re-assessed, and we feel that this concept does not appear in this portion of the text.